# Accelerated Vertical Federated Adversarial Learning through Decoupling Layer-Wise Dependencies

**Tianxing Man**[1]* **Yu Bai**[1]* **Ganyu Wang**[2] **Jinjie Fang**[1]
**Haoran Fang**[1] **Bin Gu**[1]† **Yi Chang**[1,3,4]†

[1]School of Artificial Intelligence, Jilin University, China
[2]Western University, Canada
[3]International Center of Future Science, Jilin University, China
[4]Engineering Research Center of Knowledge-Driven Human-Machine Intelligence, MOE, China

{mantx, gubin, yichang}@jlu.edu.cn {gwang382}@uwo.ca
{baiy23, fangjj24, fanghr24}@mails.jlu.edu.cn

## Abstract

Vertical Federated Learning (VFL) enables participants to collaboratively train models on aligned samples while keeping their heterogeneous features private and distributed. Despite their utility, VFL models remain vulnerable to adversarial attacks during inference. Adversarial Training (AT), which generates adversarial examples at each training iteration, stands as the most effective defense for improving model robustness. However, applying AT in VFL settings (VFAL) faces significant computational efficiency challenges, as the distributed training framework necessitates iterative propagations across participants. To this end, we propose *DecVFAL* framework, which substantially accelerates *VFAL* training through a dual-level *Dec*oupling mechanism applied during adversarial sample generation. Specifically, we first decouple the bottom modules of clients (directly responsible for adversarial updates) from the remaining networks, enabling efficient *lazy sequential propagations* that reduce communication frequency through delayed gradients. We further introduce *decoupled parallel backpropagation* to accelerate delayed gradient computation by eliminating idle waiting through parallel processing across modules. Additionally, we are the first to establish convergence analysis for VFAL, rigorously characterizing how our decoupling mechanism interacts with existing VFL dynamics, and prove that *DecVFAL* achieves an $\mathcal{O}(1/\sqrt{K})$ convergence rate matching that of standard VFLs. Experimental results show that *DecVFAL* ensures competitive robustness while significantly achieving about $3 \sim 10 \times$ speed up.

## 1 Introduction

Federated learning (FL) enables collaborative training of deep learning models among distributed participants without sharing raw data [51]. Most FL research considers Horizontal Federated Learning (HFL), which assumes distributed clients possess data with identical features but varying sample spaces [80]. **Vertical Federated Learning (VFL)** assumes distributed clients share the same samples but have different features [44, 66]. VFL model comprises a server-maintained top model and client-side bottom models that map local data features to embeddings. During inference, each client computes the local embeddings and uploads to the server through a communication channel for prediction [44]. Due to its advantages in facilitating data collaboration across multiple industries,

---

*Equal Contribution.
†Corresponding Authors.

39th Conference on Neural Information Processing Systems (NeurIPS 2025).

VFL has gained increasing attention in various domains such as recommendation systems [13, 75], finance [46, 7], healthcare [58, 6], and emerging applications [60, 19].

**Threat model.** Machine Learning (ML) models have demonstrated vulnerability to **adversarial attacks**, carefully crafted inputs designed to induce misclassification *during inference*. Distributed deployment inherently exposes VFL model to additional security threats [24, 16]. Adversarial attacks in VFL can manifest in multiple forms during inference: via third-party adversary intercepting and altering embeddings during client-server communication [16], or through malicious or colluding clients perturbing local features of raw data [52, 53]. The details of the threat model are presented in Appendix A.3. These diverse attacks underscore the unique security challenges in VFL systems, motivating the urgent need to address the VFL robustness problem.

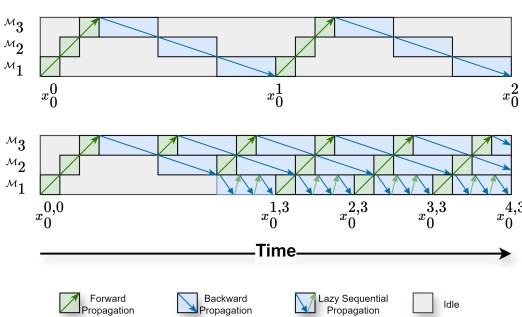

Extensive research has been conducted on defenses against adversarial attacks, with **Adversarial Training (AT)** emerging as the most empirically robust approach to date [61]. AT is a min-max robust training method that minimizes the worst-case training loss at generated adversarial examples [49]. The objective function of AT is defined in Eq. (3.1). The deployment of AT in the FL, termed Federated Adversarial Learning (FAL), has garnered attention [40]. In HFL where participants maintain complete model copies [40], studies integrate AT into clients' local training [40, 14, 1, 83, 77]. However, for applying AT in VFL settings **(VFAL)**, models are partitioned across server and clients, requiring multiple communications for adversarial sample generation—an aspect that has received limited research attention.

Figure 1: Comparison of computation efficiency for adversarial sample updates: VFL with PGD (up) versus DecVFAL (down). $\mathcal{M}_1, \mathcal{M}_2, \mathcal{M}_3$ are the layer-wise modules of VFL model. DecVFAL updates examples $4 \times 3$ times in approximately the same time as performing 2 PGD updates.

**Observation.**    As shown in Figure 1-up, inherent sequential dependencies across layers cause modules to remain idle until receiving necessary information (embeddings or gradients) from adjacent modules. Assuming $\mathcal{M}_3$ is server model and $\mathcal{M}_1, \mathcal{M}_2$ are the modules of client models, VFAL can exhibit more substantial idle waiting due to frequent server-client communications. Consequently, the training time for VFAL significantly exceeds that of regular VFLs. To illustrate (Figure 2-(a), $r = 20$), VFAL using PGD-20 (Projected Gradient Descent) requires about 20 times more computational cost than regular VFL due to 20 iterations needed to generate adversarial example. Taking into account this, a natural question arises: In light of the intensive adversarial sample generation and inherent sequential dependencies cross participants, *how can we accelerate VFAL training while maintaining robust performance?*

To tackle the computational efficiency challenge in training robust VFL models, we propose ***DecVFAL***, an accelerated **VFAL** framework through a novel **Dec**oupled backpropagation incorporating a *dual-level decoupled mechanism* for adversarial sample generation (Figure 2-(b)). DecVFAL first decouples the bottom module $f_{\mathcal{M}_1}$ (responsible for adversarial updates) from the remaining network, enabling *lazy sequential backpropagation*. This approach fixes the partial derivatives ($\delta_{f_{\mathcal{M}_1}}$) while utilizing current gradients at the bottom module ($\nabla_\eta f_{\mathcal{M}_1}$) to perform multiple sequential sample updates without requiring frequent client-server communications. Simultaneously, DecVFAL introduces *decoupled parallel backpropagation* to update partial derivatives in the remaining modules asynchronously using module-wise delayed gradients, eliminating idle waiting times and enabling truly parallel computation across all modules. Furthermore, by formulating AT as a dynamical system [39], we provide the convergence analysis for DecVFAL, accounting for multi-source approximation gradients from decoupling mechanisms and VFL architecture.

**Contributions.** In summary, the contributions of our paper are:

- We propose *DecVFAL*, a novel dual-level decoupling framework that comprehensively addresses VFAL's computational bottlenecks. DecVFAL reduces communication complexity from $r (\approx M \times N)$ to $M$ rounds while eliminating idle waiting through asynchronous parallel processing, achieving substantial speedup without sacrificing robustness.

- We provide the first convergence analysis for VFAL, overcoming the challenge of handling multi-source approximation gradients from both decoupling mechanisms and VFL framework. We rigorously prove that DecVFAL achieves an $\mathcal{O}(1/\sqrt{\mathcal{K}})$ convergence rate matching standard VFL.

- Experimental evaluation on multiple datasets demonstrates that DecVFAL achieves $3 \sim 10\times$ speedup compared to existing VFAL methods while maintaining competitive robust accuracy, making robust VFL training practically feasible for real-world deployment.

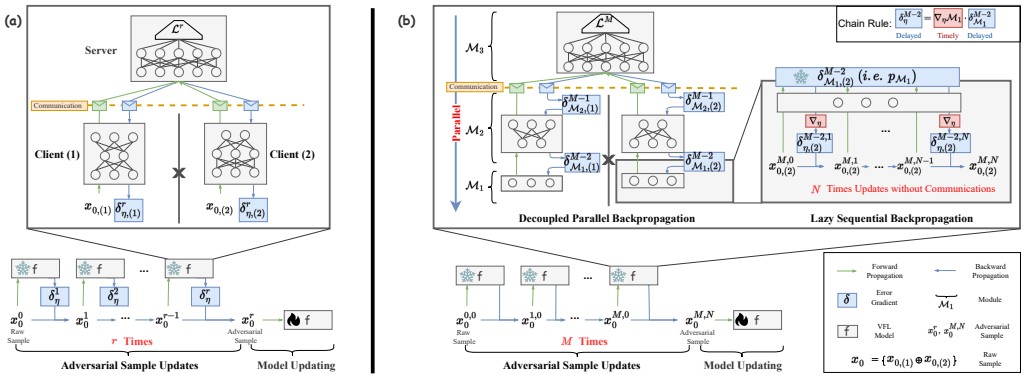

Figure 2: The conceptual comparison between VFAL with PGD (a) and DecVFAL (b): DecVFAL accesses the data $M \times N$ times with only $M$ server-client communications, each backpropagation is accelerated through the decoupling mechanism. In contrast, PGD necessitates $r$ full backpropagations to perform $r$ adversarial sample updates. ($\delta_{\mathcal{M}_k}$ represents the error gradient $\delta_{f_{\mathcal{M}_k}}$ for simplicity).

## 2 Related Works

**Accelerating Adversarial Training.** PGD-based AT methods [49] demonstrate superior defensive capabilities but present significant computational challenges. Acceleration approaches include FreeAT [55], which combines model and adversarial updates in a single backward pass; FreeLB [82], which accumulates gradients before parameter updates; and YOPO [76], which restricts updates to the first layer. Alternative FGSM-based acceleration methods like ATAS [70] and FastAT [25] offer computational advantages but struggle to achieve optimal robustness due to their reliance on single gradient steps. Importantly, these existing solutions target centralized training paradigms and fail to address the unique computational efficiency challenges in cross-participant VFAL.

**Decouple Training.** Decoupled training has emerged as effective methods for improving computational efficiency by addressing the inherently sequential propagation in neural networks. ADMM [59] decomposes optimization into parallel subproblems. Synthetic Gradients [28] enable asynchronous updates through gradient prediction. Delayed Gradient Method [27, 26, 81] temporal shifts in gradient computation for parallel processing and Lifted Machines [20, 36] transform network architecture to enhance parallelization. Notably, these techniques have yet to be applied to adversarial example generation, which is the most computationally intensive component of AT.

## 3 Preliminaries

**Notations.** A standard synchronous VFL framework comprising one central server and $C$ clients [44]. We consider a classifier implemented as a $T$-layer deep neural network $f(\Theta; x)$, where $x$ represents the input features and $\Theta$ denotes the set of trainable parameters. The training dataset is denoted as $\{(x_{0,i}, y_i)\}_{i=1}^{\mathcal{S}}$, with $\mathcal{S}$ representing the total number of samples. Each sample is composed of features from different clients, specifically $x_{0,i} = [x_{0,i,(1)}, \ldots, x_{0,i,(C)}]$. The classifier comprises client models $[f_{(1)}, \ldots, f_{(C)}]$ parameterized by $[\theta_{(1)}, \ldots, \theta_{(C)}]$ and a server model $f_s$ parameterized by $\psi$. The classifier function is expressed as $f(\Theta, x_{0,i}) = f_s\{\psi; f_{(1)}[\theta_{(1)}; x_{0,i,(1)}], \ldots, f_{(C)}[\theta_{(C)}; x_{0,i,(C)}]\}$. We give the list of notations in Table B.1.

**Adversarial Training.** AT optimizes the parameters $\Theta$ of a neural network $f$ on adversarially perturbed inputs, thereby increasing robustness against test-time attacks [49]. This process involves solving a minimax problem on the cross-entropy loss:

$$\min_{\Theta} \max_{\|\eta_i\|_\infty \leq \epsilon} \sum_{i=1}^{\mathcal{S}} \mathcal{L}(f(\Theta, x_i + \eta_i); y_i), \tag{3.1}$$

where $\mathcal{L}(\cdot; y)$ is the loss function, $\eta_i$ represents the adversarial perturbation, bounded by a non-negative scalar value $\epsilon$ that constrains the perturbation's magnitude.

**Definition 1 (Vertical Federated Adversarial Learning).** Building upon the standard VFL models and the minimax problem in AT, a $T$-layer neural network $f$ is defined recursively as: $x_t = f_t(x_{t-1}, \Theta_t), t = 1, \ldots, T$, where $x_t$ are the output of the $t$-th layer, $\Theta_t$ are the parameters of layer $f_t$, $\Theta$ denotes the concatenation of $(\Theta_t)_{1 \leq t \leq T}$. VFAL addresses the following problem:

$$\min_{\Theta} \max_{\|\eta_i\|_\infty \leq \epsilon} \quad \sum_{i=1}^{\mathcal{S}} \mathcal{L}(x_{T,i}; y_i) + \sum_{i=1}^{\mathcal{S}} \sum_{t=1}^{T} R_t(\Theta_t, x_{t-1,i}),$$
$$\text{subject to} \quad x_{1,i} = f_1(\Theta_1, x_{0,i} + \eta_i), \quad x_{t,i} = f_t(\Theta_t, x_{t-1,i}). \tag{3.2}$$

where $x_{T,i}$ is the output of the last layer $f_T$: $x_{T,i} = f(\Theta; x_{0,i} + \eta_i) = f_T(\Theta_T; f_{T-1}(\Theta_{T-1}; \ldots f_1(\Theta_1; x_{0,i} + \eta_i) \ldots))$, $R_t$ is a potential regularization term for layer $f_t$. Assume $t_c$ is the number of client model layers, for $t_c < t \leq T$, $\Theta_t = \psi_{t-t_c}$ are the server model parameters; for $0 < t \leq t_c$, $\Theta_t = [\theta_{t,(1)}, \ldots, \theta_{t,(C)}]$ are the client model parameters. $\eta_i = \eta_{i,(1)}, \ldots, \eta_{i,(C)}$ represents adversarial perturbations on sample $i$.

**Backward Locking.** In VFAL's distributed setting, a $T$-layer neural network is partitioned into $\mathcal{M}_K$ modules. The partial derivatives computation in module $f_{\mathcal{M}_k}$ depends on error gradients from module $f_{\mathcal{M}_{k+1}}$. This creates a "lock" preventing modules from updating until receiving backward results from dependent modules. As shown in Figure 2-(a), each adversarial example update in VFAL requires sequential gradient propagation from output to input layer, significantly increasing computational overhead due to cross-participant communication.

# 4 Methods

To address the training efficiency bottleneck, DecVFAL introduces a dual-level decoupled mechanism that utilizes module-wise staleness to untether the dependencies across layers inherent in VFAL. As shown in Figure 2-(b), DecVFAL utilizes delayed gradients to eliminate backward locking, enabling module-wise asynchronous backpropagation. It restricts perturbation update propagations to the bottom model to reduce full propagations and utilizes gradients from disparate iterations to achieve parallel backward computation.

## 4.1 Lazy Sequential Backpropagation

A key observation in VFAL is that the adversarial perturbation is directly coupled with the bottom module of the network. This insight allows us to decouple the bottom module $f_{\mathcal{M}_1}$ and the remaining modules $f_{\tilde{\mathcal{M}}_1}(\Theta_{\tilde{\mathcal{M}}_1}; x_{\mathcal{M}_1})$, where $f_{\tilde{\mathcal{M}}_1} = f_{\mathcal{M}_2} \circ f_{\mathcal{M}_3} \circ \ldots f_{\mathcal{M}_K}$, and $x_{\mathcal{M}_1}$ is the output of bottom module. The VFAL classifier can be rewritten as: $f(\Theta; x_0 + \eta) = f_{\tilde{\mathcal{M}}_1}(\Theta_{\tilde{\mathcal{M}}_1}; f_{\mathcal{M}_1}(\Theta_{\mathcal{M}_1}, x_0 + \eta))$. PGD-based AT (PGD-r) involves $r$ sweeps of forward and backward propagation to generate an adversarial example, resulting in extensive computational cost. To mitigate this, we introduce a "lazy" backpropagation mechanism by freezing the partial derivatives $\delta_{f_{\mathcal{M}_1}}$ of the remaining modules as a slack variable $p_{\mathcal{M}_1}$ [3]:

$$p_{\mathcal{M}_1} = \nabla_{f_{\tilde{\mathcal{M}}_1}} \left( \mathcal{L}(f_{\tilde{\mathcal{M}}_1}(f_{\mathcal{M}_1}(\Theta_{\mathcal{M}_1}; x_0 + \eta)), y) \right) \cdot \nabla_{f_{\mathcal{M}_1}} \left( f_{\tilde{\Theta}_{\mathcal{M}_1}}(f_{\mathcal{M}_1}(\Theta_{\mathcal{M}_1}; x_0 + \eta)) \right), \tag{4.1}$$

where $p_{\mathcal{M}_1}$ is obtained after each full backpropagation. The perturbation $\eta$ is updated using $p_{\mathcal{M}_1}$ and $N$-step gradient ascent, while keeping the network parameters $\Theta$ fixed (lines 7-11 in Algorithm 1). As shown in Figure 1, DecVFAL accesses the data $M \times N$ times for each adversarial example generation while only requiring $M$ full forward and backward propagation, where $M \ll r$.

This frozen slack variable introduces an oracle error in adversary updating, resulting in a delayed gradient. Inspired by the optimal control theory [38, 39, 54] and under **Assumptions** in (B.2), we bound costate difference at bottom module in Lemma 1.

**Lemma 1.** **Bound the costate/gradient difference at the bottom module.** *There exists a constant $G'$ dependent on $T$ and $K$ such that for all $n \in \{0, \ldots, N\}$, $m \in \{0, \ldots, M\}$, and $i \in \{1, \ldots, S\}$:*

$$\left\| p_{\mathcal{M}_1, i}^{m-\tau_1, 0} - p_{\mathcal{M}_1, i}^{m, N} \right\| \leq G' \alpha_\eta (\mathcal{M}_K N - 1), \tag{4.2}$$

---

[3] This is inspired by YOPO [76], but extends beyond first-layer limitations to support multi-layer modules, offering greater flexibility of network partitioning in VFAL settings.

where $G' = TK^{T+1}(K^T + T(T-1)K^{2T-2} + TK^{2T})$, $p_{\mathcal{M}_1,i}^{m-\tau_1,0}$ is the delayed gradient to the first module $\mathcal{M}_1$ at iteration $m$, $\tau_1$ is the delay of module $\mathcal{M}_1$ raised from parallel backpropagation.

## 4.2 Decoupled Parallel Backpropagation

We decouple backpropagation across the entire network using delayed gradients, enabling parallel updates of the partial derivatives in the remaining modules. The forward pass is performed from module 1 to module $\mathcal{M}_K$. In backward pass, all modules except the last one store delayed error gradients to perform the backward computation without locking. The module $f_{\mathcal{M}_k}$ keeps the stale error gradient $\frac{\delta \mathcal{L}^{m-\tau_k}}{\delta x_{\mathcal{M}_k}}$, $\tau_k = \mathcal{M}_K - \mathcal{M}_k$. While lazy backpropagation in the bottom modules, backward computation in the remaining modules $f_{\mathcal{M}_k}$ is:

$$\frac{\delta \mathcal{L}^{m-\tau_k}}{\delta x_{t-1}} = \frac{\delta x_t}{\delta x_{t-1}} \frac{\delta \mathcal{L}^{m-\tau_k}}{\delta x_t}, t \in (t_{\mathcal{M}_{k-1}}, t_{\mathcal{M}_k}]. \tag{4.3}$$

Each module receives a gradient from its dependent module for subsequent computation. The gradients across modules exhibit varying time delays. Moving from module 1 to $\mathcal{M}_K$, the corresponding time delays $\tau_k$ decrease from $\mathcal{M}_K - 1$ to 0, delay 0 indicates up-to-date gradients. This mechanism breaks the backward locking constraint, enabling decoupled parallel backpropagation.

For private guarantee[4], We implement DecVFAL within VFL-CZOFO, a hybrid cascaded VFL framework [62] with Zeroth Order Optimization (ZOO) and compression, and analyze the errors caused by *multi-source approximate gradients* due to existing VFL techniques and DecVFAL in Lemma 2.

**Lemma 2.** ***Bound the gradient to*** $\eta$***.*** *Under hybrid cascaded VFL, the gradient $\nabla_\eta \mathcal{A}$ respect to $\eta$ involves estimation gradient $\nabla_\eta \hat{\mathcal{A}}$ from ZOO (Appendix A.5) and compression gradient $\hat{\nabla}_\eta \mathcal{A}$ (Appendix A.6). Under the **Assumption 1**, and **Lemma 3, 5**, at the $i$-th sample and $k$-th iteration, the pseudo-partial derivative for $\eta$ satisfies the following inequality $\hat{\eta}_i = \underset{\substack{m=1,\ldots,M \\ n=1,\ldots,N}}{argmin} \left\| \hat{\nabla}_\eta \hat{\mathcal{A}}_i \left( \eta_i^{m,n}, \psi_i, \theta_i \right) \right\|$, we define $G = KG'$, $\alpha_x < \frac{1}{L_{\eta\eta}}$ then:*

$$\mathbb{E} \left\| \hat{\nabla}_\eta \hat{\mathcal{A}}_i(\hat{\eta}_i, \psi_i, \theta_i) \right\|^2$$
$$\leq \left[ D(\mathcal{X}) L_{\eta\eta}^2 \left( 1 - \frac{z}{L_{\eta\eta}} \right)^{MN+1} + \frac{2G^2}{L_{\eta\eta}} (\mathcal{M}_K N - 1)^2 \left( \frac{2}{z} + \frac{1}{2L_{\eta\eta}} \right) \right] \times 3 \left( H_\theta^2 C \mathcal{E}^k + \frac{L^2 \mu^2 d^2}{4} + K^2 \right). \tag{4.4}$$

---

**Algorithm 1:** DecVFAL algorithm

---

**Input:** Learning rates $\alpha_\eta, \alpha_\psi, \alpha_\theta$; Train set $\{X, Y\}$.
**Output:** Model parameters $\Theta = \{\theta_{(1)}, \theta_{(2)}, \ldots, \theta_{(C)}, \psi\}$.

1 **Initialization:** Clients and Server initialize model parameters $\theta_{(1)}, \theta_{(2)}, \ldots, \theta_{(C)}, \psi$.
2 **while** *not convergent* **do**
3      Randomly select a sample $x$;
4      **for** $m = 1$ *to* $M$ **do**
5          $\mathcal{L}^m \leftarrow f(x_0 + \eta^{m,n})$;
6

             **for** *k = 1 to* $\mathcal{M}_K$ ***in parallel*** **do**          **Decoupled Parallel Backpropagation**
                 **if** *k = 1* **then**
                     **for** *n = 0 to N-1* **do**      **Lazy Sequential Propagation**
                         $x_{\mathcal{M}_1}^{m,n} \leftarrow f_{\mathcal{M}_1}(x_0 + \eta^{m,n})$;
                         Updates adversarial perturbation: $\eta^{m,n+1} \leftarrow \eta^{m,n} + \alpha_\eta p_{\mathcal{M}_1} \nabla_\eta f_{\mathcal{M}_1}$;

                 Backward computation with delayed gradient:
                   $\frac{\delta \mathcal{L}^{m-\tau_k}}{\delta x_{t-1}} \leftarrow \frac{\delta x_t}{\delta x_{t-1}} \frac{\delta \mathcal{L}^{m-\tau_k}}{\delta x_t}, t \in (t_{\mathcal{M}_{k-1}}, t_{\mathcal{M}_k}]$;
7

8      **for** *each client c* **do**
9          Update client model parameters: $\theta_{(c)}^{k+1} \leftarrow \theta_{(c)}^k - \alpha_\theta \nabla_\theta \mathcal{L}(f(x_0 + \eta^{m,n}))$;
10      Update server model parameters: $\psi^{k+1} \leftarrow \psi^k - \alpha_\psi \nabla_\psi \mathcal{L}(f(x_0 + \eta^{m,n}))$;

---

[4]According to [62], due to the introduction of ZOO, DecVFAL satisfies differential privacy guarantee.

### 4.3 Algorithm

Algorithm 1 implements DecVFAL framework within the VFL setting where clients and server collaboratively train a robust model. In each iteration, after randomly selecting a sample $x$, the framework performs $M$ full propagations where all modules across clients and server operate in parallel. For each module $\mathcal{M}_k$, partial derivatives are computed using delayed gradients $\delta\mathcal{L}^{m-\tau_k}/\delta x_t$ with delay $\tau_k = \mathcal{M}_K - \mathcal{M}_k$. Simultaneously, the bottom module (typically at client side) performs $N$ adversarial perturbation updates through lazy sequential propagation: $\eta^{m,n+1} \leftarrow \eta^{m,n} + \alpha_\eta p_{\mathcal{M}_1} \nabla_\eta f_{\mathcal{M}_1}$, using a fixed gradient $p_{\mathcal{M}_1}$ without requiring additional client-server communications. After adversarial example generation, clients update their parameters $\theta_{(c)}$ and the server updates its parameter $\psi$, completing one training iteration.

### 4.4 Acceleration of DecVFAL

DecVFAL accelerates VFAL training through our dual-level decoupling mechanism. The lazy sequential back-propagation enables $M \times N$ adversarial sample updates while requiring only $M$ full propagations, empirically achieving comparable robustness to PGD-$r$ when $M \times N$ is slightly larger than $r$. For a standard full propagation taking time $\mathcal{T}$, our decoupled parallel backpropagation reduces this to $\frac{\mathcal{T}}{\mathcal{M}_K}$ by eliminating idle waiting. Unlike prior research [27, 26] that uses delayed gradients solely for model training (limiting acceleration to $\mathcal{T}_{for} + \frac{\mathcal{T}_{back}}{\mathcal{M}_K}$), our method enables concurrent forward and backward propagation during adversarial sample generation since parameters remain constant. Ideally, this further reduces computation time to $\frac{\mathcal{T}}{\mathcal{M}_K}$.

## 5 Convergence Analysis

**Assumptions.** The formal definition and detailed discussion of the assumptions are in Appendix B.2. We make several crucial assumptions: the functions $f_t$, $f_c$, $\mathcal{L}$, and $R_t$ are $K$-Lipschitz continuous in $x$, uniformly with respect to $\theta$ and $\psi$, the gradient of the adversarial loss function, $\nabla\mathcal{A}_i(\eta, \psi, \theta)$, satisfies Lipschitz conditions (Assumption 1); the adversarial loss function $\mathcal{A}_i(\eta, \psi, \theta)$ possesses an unbiased gradient (Assumption 2) and is characterized by bounded Hessian matrices $H_\psi$ and $H_\theta$ (Assumption 3), as well as bounded block-coordinate gradients $Q_\psi$ and $Q_\theta$ (Assumption 4); $\mathcal{A}_i(\eta, \psi, \theta)$ exhibits $z$-strong concavity with respect to $\eta$ (Assumption 5).

**Theorem 1.** Under Assumptions (1–5), if the step sizes satisfy $\alpha_\eta < 1/L_{\eta\eta}$, $\alpha_m = \min\{\alpha_\psi, \alpha_\theta\}$, $\alpha_M = \max\{\alpha_\psi, \alpha_\theta\}$, $\frac{\alpha_M}{\alpha_m} < \infty$, with $\eta_i^* = \mathrm{argmax}_\eta \mathcal{A}_i(\eta, \psi, \theta)$, $\Lambda = \mathcal{R}\left(\eta^{*,0}, \psi^0, \theta^0\right) - \inf_k(\mathcal{R}\left(\eta^{*,k}, \psi^k, \theta^k\right))$, $L_\star = \max\{L, L_\psi, L_\theta, L_{\psi\eta}, L_{\theta\eta}\}$, $\xi_\theta = \{1 + L_\theta \alpha_M\}$, and $\xi_\psi = \{1 + L_\psi \alpha_M\}$, then:

$$\frac{1}{\mathcal{K}}\sum_{k=0}^{\mathcal{K}-1}\mathbb{E}\left[||\nabla\mathcal{R}\left(\eta^{*,k}, \psi^k, \theta^k\right)||^2\right] \leq \mathcal{I}_1 + \mathcal{I}_2 + E_p + E_c + E_z, \qquad (5.1)$$

where $\pi(M, N) = D(\mathcal{X})L_{\eta\eta}^2\left(1 - \frac{z}{L_{\eta\eta}}\right)^{MN+1} + \frac{2G^2}{L_{\eta\eta}}\left[\mathcal{M}_K N - 1\right]^2\left(\frac{2}{z} + \frac{1}{2L_{\eta\eta}}\right)$, and:

$\mathcal{I}_1 = \dfrac{2\Lambda}{\alpha_m \mathcal{K}}$   (first-order optimization convergence term),

$\mathcal{I}_2 = \dfrac{2L_\star \alpha_M^2(\sigma_\psi^2 + \sigma_\theta^2)}{\alpha_m}$   (stochastic gradient descent term),

$E_z = \mu^2\left[\dfrac{3\alpha_M \xi_\theta L_\star^2 d^2}{4\alpha_m} + \dfrac{3\pi(M, N)L_\star^2 d^2 a_M(\xi_\psi/2 + 3\xi_\theta/4)}{a_m z^2}\right]$   (zeroth-order error term),

$E_c = \mathcal{E}\dfrac{\alpha_M}{\alpha_m}\left(2\xi_\psi H_\psi^2 C + 3\xi_\theta Q_\theta^2 H_\theta^2 C + \dfrac{3H_\theta^2 C\pi(M, N)(2\xi_\psi + 3\xi_\theta)L_\star^2}{z^2}\right)$   (compression error term),

$E_p = \dfrac{3\alpha_M K^2 \pi(M, N)}{\alpha_m z^2}(2\xi_\psi + 3\xi_\theta)L_\star^2$   (decoupled backpropagation error term).

**Corollary 1.** *If we choose $\alpha_\theta$ and $\alpha_\psi$ as $\frac{1}{\sqrt{\mathcal{K}}}$, $\mu = \frac{1}{\mathcal{K}^{\frac{1}{4}}}$, $\mathcal{E} = \mathcal{O}(\frac{1}{\sqrt{\mathcal{K}}})$, $\Gamma = \mathcal{O}(\frac{1}{\sqrt{\mathcal{K}}})$, we derive the sublinear convergence rate:*

$$\frac{1}{\mathcal{K}}\sum_{k=0}^{\mathcal{K}-1}\mathbb{E}\left[||\nabla\mathcal{R}\left(\eta^{*,k}, \psi^k, \theta^k\right)||^2\right] \leq \mathcal{O}(\frac{1}{\sqrt{\mathcal{K}}}) + \mathcal{O}(\frac{N\mathcal{M}_K}{M}). \qquad (5.2)$$

**Remark 1.** *The bias term $\mathcal{O}(\frac{N\mathcal{M}_K}{M})$ reveals a direct trade-off between model modularity and performance. As the number of modules $\mathcal{M}_K$ increases, the bias grows proportionally, which aligns with our empirical observations in Table 7.*

**Remark 2.** *For fixed $\mathcal{M}_K$, the bias term exhibits properties analogous to those identified in [54]. $\pi(M, N)$ is monotonically decreasing with respect to $M$ and convex with respect to $N$. This functional behavior implies an*

*optimal strategy: maximizing $M$ subject to communication budget, while carefully calibrating $N$ to the critical point where marginal returns diminish. Our ablation studies (Figure 5)validate these theoretical findings.*

**Proof Sketch.** Our proof begins by recasting the original min-max optimization problem as a Hamiltonian system (Appendix A.4). The convergence analysis hinges on three distinct types of approximate gradients: delayed gradient (Lemma 1 and Lemma 2), compression gradient (Lemma 5), and estimated gradient (Lemma 3). We establish the global convergence of the framework by proving that the loss function $\mathcal{L}(\eta, \psi, \theta)$ is L-smooth (Assumption 1). Through a comprehensive analysis of the $M$ loop, $N$ loop, and outer loop dynamics, we prove the asymptotic convergence of model parameters (Theorem 1). The complete convergence analysis of DecVFAL, with detailed proofs, is presented in Appendix B.

## 6 Experiments

We extensively evaluated DecVFAL against diverse baseline VFAL methods. Ablation studies were conducted to analyze component contributions. Detailed procedures are available in Appendix C, and the code is accessible at https://github.com/workelaina/DecVFAL.git.

### 6.1 Experiment Setups

**Datasets.** Real-world VFL datasets are proprietary and not publicly accessible. Therefore, we utilized three public datasets instead for our main experiments: MNIST [34], CIFAR-10 [31], large scale dataset CIFAR-100 [31], and Tiny-ImageNet [33] (Appendix C.6). These datasets were vertically partitioned among all participants, with each client retaining a portion of features for each sample, while the server exclusively held the labels. Detailed information about the dataset partitioning can be found in Section C.1.

**Baselines.** We deploy the baseline algorithms and DecVFAL in a hybrid cascaded VFL framework, synchronous VFL-CZOFO [62]. The implemented AT algorithms include PGD-$r$ [49], and the accelerated AT methods: FreeAT-$r$ [55], FreeLB-$r$ [82], and YOPO-$m$-$n$ [76]. Additionally, we integrated data parallelism, model parallelism, and asynchronous mechanisms with PGD, resulting in DP-PGD, MP-PGD, and Asy-PGD, respectively. We also compare with FGSM-based accelerated AT methods, ATAS [70], FastAT [25].

**Adversarial attack.** Following the threat model in VFL (Appendix A.3), we employ various adversarial attack methods including FGSM [32], PGD-$r$ [49], and CW [4]. We also simulate scenarios where malicious clients cannot directly obtain gradients and implement black-box attacks: CERTIFY (CER) [11], zero-order-based FGSM (ZO-FGSM) and PGD (ZO-PGD) [8]. Additionally, Considering the case of the third-party adversary, we employ adversarial attacks that corrupt embeddings using different corrupted client selection methods: Thompson Sampling with Empirical Maximum Reward (E-TS) [16] and All Corruption Pattern (ALL).

**Training procedures.** We deployed a VFL setup with one server and two clients for the experiments. For the MNIST dataset, each client used a two-layer perceptron [5] and the server employed a single-layer perceptron, with the model partitioned into three modules and trained using batch size 32. On CIFAR-10, while the server maintained a single-layer perceptron, each client utilized ResNet-18 [6] divided into two modules (first layer and remaining layers), creating a three-module structure across participants with batch size 80. For CIFAR-100, we used ResNeXt-50 on client sides, similarly partitioned into three modules. Additional experiments with ResNet-18 on MNIST followed the same partitioning scheme as CIFAR-10. All models were trained to convergence using Adam optimizer with a fixed learning rate of 0.0001 for fair comparison. Detailed parameter settings and hardware specifications for the training procedures are summarized in Appendix C.3 and Table 13.

### 6.2 Evaluation on computational efficiency

For each dataset, we trained models to converge and plotted training and testing curves in Figures 3 and 4. DecVFAL achieved better test accuracy than other baseline algorithms in significantly less time on MNIST. Due to setting close parameters to specify the number of full propagations (Table 11) for CIFAR10, DecVFAL achieved a convergence speed comparable to FreeAT and FreeLB, while delivering better robustness, as shown in Table 2.

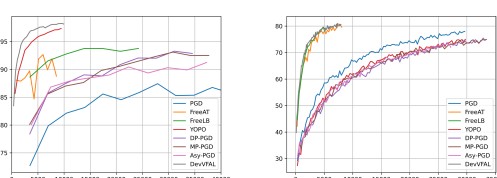

Figure 3: MNIST training-testing curves

Figure 4: CIFAR-10 training-testing curves

---

[5]For different model architectures, we conduct experiments with ResNet on MNIST (Appendix C.5).

[6]Residual connections naturally align with dynamical systems theory [9, 68], where entire residual blocks function as single dynamical units.

## 6.3 Evaluation on Robustness

For the MNIST dataset, DecVFAL demonstrates the most optimal trade-off between computational efficiency and model robustness. As shown in Table 1, DecVFAL achieves the best robust performance while requiring only 1/10 of the training time per epoch compared to standard PGD adversarial training. Similarly for CIFAR-10, the results in Table 2 show that DecVFAL achieves comparable robust performance under most adversarial attacks while requiring only 1/3 of the training time per epoch compared to PGD. These results consistently demonstrate the effectiveness of DecVFAL in balancing computational efficiency with model robustness.

Table 1: Results of MNIST Robust Training

| Training Methods | Clean Accuracy | White-Box Adv. Atk | | | Black-Box Adv. Atk | | | Third Adv. | | Train Time (s/epoch) |
|---|---|---|---|---|---|---|---|---|---|---|
| | | FGSM | PGD | CW | CER | ZO-FGSM | ZO-PGD | ALL | E-TS | |
| None | **96.46** | 47.75 | 8.58 | 56.75 | 56.37 | 55.97 | 60.44 | 36.01 | 40.27 | **106.86** |
| PGD | 92.31 | 74.90 | 57.85 | 90.09 | 88.92 | 87.30 | 83.37 | 36.71 | 41.23 | 3484.57 |
| FreeAT | 92.68 | 67.29 | 41.33 | 85.11 | 84.13 | 83.51 | 80.18 | 19.01 | 20.82 | 853.64 |
| FreeLB | 93.77 | 57.18 | 18.76 | 85.30 | 82.47 | 82.30 | 79.73 | 65.33 | 71.11 | 3459.81 |
| YOPO | 96.13 | 86.36 | 73.52 | 92.49 | 91.63 | 91.17 | 88.06 | 79.81 | 84.84 | 629.43 |
| DP-PGD | 93.28 | 78.64 | 60.97 | 88.40 | 86.60 | 86.49 | 82.84 | 51.72 | 56.68 | 3451.44 |
| MP-PGD | 93.11 | 75.23 | 48.98 | 78.82 | 76.65 | 76.28 | 76.19 | 48.11 | 54.67 | 3423.91 |
| Asy-PGD | 91.25 | 72.40 | 50.41 | 84.53 | 82.55 | 82.10 | 79.50 | 38.42 | 42.99 | 3724.47 |
| NoLazy | 90.17 | 72.58 | 55.53 | 89.76 | 85.32 | 77.03 | 50.96 | 26.66 | 26.05 | **317.78** |
| DecVFAL | **96.30** | **91.62** | **77.68** | **92.84** | **91.91** | **92.13** | **89.21** | **92.20** | **94.53** | 355.16 |

Table 2: Results of CIFAR-10 Robust Training

| Training Methods | Clean Accuracy | White-Box Adv. Atk | | | Black-Box Adv. Atk | | | Third Adv. | | Train Time (s/epoch) |
|---|---|---|---|---|---|---|---|---|---|---|
| | | FGSM | PGD | CW | CER | ZO-FGSM | ZO-PGD | ALL | E-TS | |
| None | **83.93** | 53.32 | 55.42 | 62.59 | 50.39 | 52.38 | 55.58 | 76.06 | **78.93** | **70.03** |
| PGD | 78.00 | 59.08 | 68.47 | 76.73 | 70.00 | 70.32 | 70.56 | 69.54 | 72.67 | 296.35 |
| FreeAT | 80.09 | 63.63 | 61.93 | **77.01** | 68.99 | 70.99 | 71.85 | 71.44 | 74.86 | 252.11 |
| FreeLB | 81.58 | 52.09 | 54.91 | 63.70 | 53.91 | 56.92 | 59.17 | **76.30** | 78.70 | 301.43 |
| YOPO | 75.34 | 58.80 | 68.11 | 74.68 | 70.10 | 69.97 | 69.96 | 64.38 | 69.05 | 297.45 |
| DP-PGD | 75.47 | 59.37 | 68.24 | 74.56 | 69.79 | 69.74 | 70.04 | 66.19 | 69.42 | 331.93 |
| MP-PGD | 74.92 | 59.38 | 68.14 | 74.30 | 69.92 | 69.53 | 69.90 | 64.70 | 68.66 | 334.48 |
| Asy-PGD | 73.32 | 57.00 | 66.61 | 72.48 | 67.56 | 67.93 | 67.83 | 63.36 | 67.83 | 331.45 |
| DecVFAL | **81.83** | **63.69** | **68.59** | 74.72 | **71.31** | **71.05** | **72.07** | 74.93 | 77.75 | **98.99** |

## 6.4 Experiments on Real-World Datasets

To further validate the generality of our framework in practical federated applications, we conducted experiments on two *real-world datasets*: COVID-19 Image Data Collection [50] and Credit Card Fraud Detection [3]. These datasets are widely used in healthcare and financial anomaly detection scenarios, where data are naturally vertically partitioned across organizations. We simulated two clients, each holding disjoint feature subsets of the same samples.

Table 3: Results of robust training on real-world datasets

| Methods | COVID-19 | | | Credit Fraud | | | Speedup |
|---|---|---|---|---|---|---|---|
| | Clean | PGD | AA | Clean | PGD | AA | vs PGD |
| None | **98.19** | 0.00 | 0.00 | **92.45** | 2.86 | 3.38 | – |
| PGD | 69.20 | **67.39** | **66.30** | 88.28 | 42.19 | **49.48** | 1.00× |
| FreeAT | 77.53 | 50.54 | 40.76 | 92.44 | 40.36 | 5.72 | 2.54× / 1.33× |
| YOPO | 77.90 | 66.49 | 58.51 | 90.10 | 32.29 | 22.39 | 3.39× / 2.00× |
| DecVFAL | 88.22 | 62.86 | 44.75 | 89.58 | **47.13** | 47.39 | **3.72× / 2.35×** |

All methods shared identical hyperparameters as in previous experiments. Table 3 summarizes the robustness and efficiency performance on both datasets. Across both settings, DecVFAL consistently delivers the best robustness–efficiency trade-off. On the COVID-19 dataset, it achieves a **3.72×** speedup relative to standard PGD while maintaining comparable robust accuracy under multiple attacks. On the Credit dataset, DecVFAL achieves a **2.35×** speedup with strong adversarial robustness, outperforming prior acceleration methods.

## 6.5 Experiments on large dataset

We further evaluate the robustness results of DecV-FAL on CIFAR-100, which is a more difficult dataset with more classes. As shown in Table 4, DecVFAL achieves the highest robust accuracy against all types of adversarial attacks, while reducing training time to only 1/3 of standard PGD. Although FreeAT and FreeLB perform slightly better on clean data, they show significant performance degradation under adversarial attacks. This degradation occurs primarily because their simultaneous updates of model parameters and adversarial samples lead to overfitting, highlighting the advantage of DecVFAL.

Table 4: Results of CIFAR-100 Robust Training

| Training | Robust Accuracy (%) | | | | | Train Time |
| Methods | Clean | FGSM | PGD | ZO-PGD | ETS | (s/epoch) |
|---|---|---|---|---|---|---|
| None | 53.55 | 16.89 | 9.84 | 27.07 | 32.51 | 123.65 |
| PGD | 51.09 | 39.46 | 39.15 | 45.40 | 27.82 | 1106.19 |
| FreeAT | 54.06 | 36.30 | 35.17 | 47.54 | 43.32 | 946.83 |
| FreeLB | **54.35** | 15.30 | 8.19 | 21.91 | 45.12 | 935.29 |
| YOPO | 51.91 | 37.44 | 36.00 | 46.02 | 31.65 | 587.70 |
| DP-PGD | 49.55 | 35.02 | 37.72 | 43.45 | 23.81 | 1098.77 |
| MP-PGD | 48.48 | 35.99 | 35.52 | 41.70 | 16.19 | 1150.38 |
| Asy-PGD | 51.19 | 39.83 | 39.36 | 45.91 | 25.00 | 1115.73 |
| DecVFAL | 54.33 | **43.43** | **43.24** | **49.60** | **49.57** | **446.26** |

## 6.6 Comparison with FGSM-based Methods

We also compared with computationally efficient FGSM-based methods (FastAT and ATAS). Table 5 presents the results across different perturbation budgets. DecVFAL-6-2 consistently achieves superior robustness while requiring only 1/3 of PGD's training time. Notably, DecVFAL-3-3 matches FastAT's training efficiency while delivering better robustness, demonstrating DecVFAL balances computational efficiency and model robustness through its configurable parameters.

Table 5: Comparison with FGSM-based methods

| Method | Time (h) | $\varepsilon = 4/255$ | | $\varepsilon = 8/255$ | | $\varepsilon = 12/255$ | |
| | | PGD-10 | AA | PGD-10 | AA | PGD-10 | AA |
|---|---|---|---|---|---|---|---|
| PGD-10 | 8.23 | 68.47 | **63.08** | 44.61 | 34.55 | 34.59 | 16.29 |
| FastAT | **1.45** | 63.61 | 57.69 | 40.41 | 30.62 | 30.58 | 14.20 |
| ATAS | 1.85 | 66.23 | 61.08 | 42.56 | 32.46 | 32.85 | 15.30 |
| DecVFAL-6-2 | 2.75 | **68.59** | 62.34 | **46.27** | **36.48** | **36.14** | **18.72** |
| DecVFAL-3-3 | **1.45** | 66.19 | 60.58 | 42.96 | 31.76 | 31.98 | 15.11 |

## 6.7 Ablation Study

**Impact of the number of clients** $C$**.**  We conducted additional experiments on the MNIST dataset by varying the number of clients $C$ among 3, 5, and 7. DecVFAL consistently achieved superior robustness and enhanced computational efficiency across all client configurations compared to baseline methods. Additionally, we tested DecVFAL and baselines against third-party attacks in a 7-client setting, including corruption pattern selection and malicious client scenarios (see Appendix 6.7). DecVFAL still demonstrated better robust performance.

Table 6: Results for different number of clients $C$

| No. | Training | Clean | White-Box Adv. Atk | | | Black-Box Adv. Atk | | | Third Adv. | Train Time |
| Clients | Methods | Accuracy | PGD | FGSM | CW | CER | ZO-FGSM | ZO-PGD | ALL | (s/epoch) |
|---|---|---|---|---|---|---|---|---|---|---|
| 3 | PGD | 98.05 | 64.56 | 82.83 | 96.02 | 96.46 | 93.98 | 94.72 | 89.56 | 1015.80 |
| 5 | PGD | 96.78 | 69.50 | 84.51 | 93.00 | 95.20 | 92.36 | 93.08 | 78.62 | 1145.63 |
| 7 | PGD | 96.18 | 63.86 | 79.96 | 90.96 | 93.30 | 90.37 | 94.12 | 69.36 | 1158.11 |
| 3 | DecVFAL | **98.67** | 80.82 | 89.50 | 97.28 | 97.90 | 96.03 | 96.97 | 93.83 | 88.29 |
| 5 | DecVFAL | 98.30 | 76.52 | 87.39 | 97.34 | 97.54 | 93.85 | 95.73 | 91.87 | 92.93 |
| 7 | DecVFAL | 96.84 | 76.57 | 87.17 | 83.90 | 96.21 | 93.23 | 90.80 | 81.21 | 94.83 |

**Impact of the number of modules** $\mathcal{M}_K$**.**  To verify the impact of the number of modules $\mathcal{M}_K$, we conducted additional experiments on the MNIST dataset. Each client employed a ResNet-18 model, which was partitioned into varying numbers of modules: 2, 3, 4, 5, and 6. All configurations were trained for an identical number of epochs. As indicated by Remark 1, increasing the number of modules leads to larger errors in the gradient of $\eta$, which in turn negatively impacts the algorithm's accuracy. This is demonstrated by the results shown in Table 7.

Table 7: Results of the diverse number of modules

| | Robust Accuracy (%) | | |
| Split Positions for Modules | Clean | FGSM | PGD |
|---|---|---|---|
| $[: 1 : 18 : 19]$ | **98.90** | **48.79** | **57.49** |
| $[: 1 : 9 : 18 : 19]$ | 98.71 | 45.88 | 55.55 |
| $[: 1 : 9 : 13 : 18 : 19]$ | 98.58 | 44.32 | 53.42 |
| $[: 1 : 5 : 9 : 13 : 18 : 19]$ | 98.69 | 47.09 | 45.49 |
| $[: 1 : 5 : 9 : 13 : 17 : 18 : 19]$ | 98.22 | 38.44 | 40.63 |

**Impact of lazy propagation.**  We evaluated the effectiveness of lazy propagation on the MNIST dataset using DecVFAL framework (M=5, N=10). By setting N=1, we implemented a variant without lazy propagation, denoted as "Nolazy". As shown in Table 1, Nolazy only updates each sample 5 times to generate adversarial examples. While the decoupled parallel propagation still provides some training time advantage, the overall model performance cannot be guaranteed under this configuration.

**Limitation of the setting of** $M$ **and** $N$**.**  We conducted experiments on the MNIST dataset to explore the dependence on parameters $M$ and $N$. Figure 5 illustrate the change in accuracy with a fixed $M = 3, 5, 8, 10$, respectively, while varying $N$. The performance exhibits a non-monotonic relationship with $N$, as analyzed in

Remark 2. Initially, performance improves as $N$ increases, however, beyond a certain threshold, it deteriorates significantly. This pattern emphasizes the critical importance of optimal $N$ selection to maintain model accuracy.

**Impact of split position.** We conducted additional experiments on MNIST to evaluate the effect of different split positions. The server model is still kept as a single-layer perceptron, while each client utilized a ResNet-18 model that is split into two modules at various positions. The results in Table 8 demonstrate that DecVFAL performs well across various split positions compared to PGD. However, as more layers are included in the bottom module during lazy sequential backpropagation, the computational load increases, leading to longer training time.

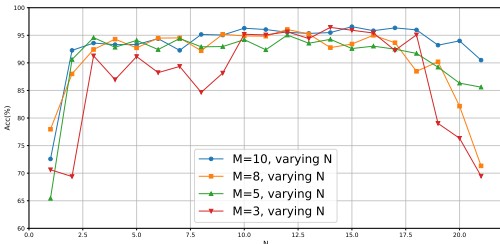

Figure 5: Result of fixed $M$ and varying $N$

Table 8: Results of different split positions

| Split Positions | Robust Accuracy (%) | | | Train Time |
|---|---|---|---|---|
| $[: \mathcal{M}_1 : \mathcal{M}_2 : \mathcal{M}_3]$ | Clean | FGSM | PGD | (s/epoch) |
| $[: 1 : 18 : 19]$ | **98.90** | **48.79** | **57.49** | **107.54** |
| $[: 5 : 18 : 19]$ | 98.77 | 43.03 | 42.98 | 226.76 |
| $[: 9 : 18 : 19]$ | 98.75 | 41.33 | 49.77 | 318.12 |
| $[: 13 : 18 : 19]$ | 98.83 | 39.73 | 43.46 | 431.14 |
| $[: 17 : 18 : 19]$ | 98.43 | 36.36 | 45.88 | 538.65 |
| PGD | 98.48 | 32.53 | 41.93 | 575.45 |

**Evaluation under attacks involving corruption pattern selection.** To further assess our framework's resilience in more complex attack scenarios, we conducted experiments on the MNIST dataset using seven clients. Specifically, we evaluated DecVFAL and baseline methods against attacks involving corruption pattern selection. In this setup, adversaries could selectively corrupt client data or communications. The server model remained a single-layer perceptron. We implemented various corruption patterns, including E-TS, RC, and FC. As shown in Table 9, the results demonstrated that even under these challenging conditions, DecVFAL maintained superior performance compared to baseline methods.

Table 9: Results of evaluation under attacks with various corruption patterns.

| | Corrupted clients: 1/7 | | | | | | | | |
|---|---|---|---|---|---|---|---|---|---|
| Training | White-Box Adv. Atk | | | Black-Box Adv. Atk | | | Third Adversary Atk | | |
| Methods | PGD | FGSM | CW | CER | ZO-FGSM | ZO-PGD | E-TS | FC | RC |
| PGD | 92.238 | 94.01 | 93.85 | 94.091 | 94.03 | 93.399 | 88.842 | 88.922 | 88.982 |
| DecVFAL | 95.613 | 96.575 | 96.795 | 96.605 | 96.585 | 96.044 | 93.048 | 93.87 | 93.219 |
| | Corrupted clients: 3/7 | | | | | | | | |
| Training | White-Box Adv. Atk | | | Black-Box Adv. Atk | | | Third Adversary Atk | | |
| Methods | PGD | FGSM | CW | CER | ZO-FGSM | ZO-PGD | E-TS | FC | RC |
| PGD | 79.888 | 87.099 | 93.359 | 94.101 | 92.819 | 92.758 | 77.364 | 78.105 | 77.754 |
| DecVFAL | 86.569 | 92.949 | 96.044 | 96.404 | 95.543 | 94.922 | 84.816 | 85.577 | 84.685 |
| | Corrupted clients: 5/7 | | | | | | | | |
| Training | White-Box Adv. Atk | | | Black-Box Adv. Atk | | | Third Adversary Atk | | |
| Methods | PGD | FGSM | CW | CER | ZO-FGSM | ZO-PGD | E-TS | FC | RC |
| PGD | 64.724 | 80.689 | 91.526 | 93.279 | 90.935 | 91.587 | 69.03 | 68.53 | 69.111 |
| DecVFAL | 78.235 | 87.31 | 91.987 | 96.044 | 93.049 | 94.121 | 75.972 | 76.062 | 76.322 |

# 7 Conclusions

This paper presented DecVFAL, a framework that significantly accelerates VFAL while maintaining robustness. DecVFAL incorporates a dual-level decoupled mechanism to enable lazy sequential and decoupled parallel backpropagation for adversarial example generation. This approach maintains the same theoretical convergence rate of $\mathcal{O}(1/\sqrt{\mathcal{K}})$ as regular VFLs while providing acceleration in each iteration, resulting in significant empirical improvements with 3-10 fold speedup across various datasets. Comprehensive experiments demonstrate DecVFAL's effectiveness across various neural architectures and VFL configurations.

# Acknowledgement

Dr. Yi Chang was supported by the National Key R&D Program of China under Grant No. 2023YFF0905400, and the National Natural Science Foundation of China through grants No. U2341229 and No. 62076138.

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

# Appendix Contents

# A  Background

## A.1  Vertical Federated Learning

VFL encompasses a range of architectural designs tailored for collaborative machine learning across multiple parties. These architectures, distinguished by data and parameter distribution, as well as the trainability of the server model, include Aggregated Vertical Federated Learning ($aggVFL$) [17, 45], where client parties contribute intermediate results aggregated through a non-trainable function in the server party; Aggregated Vertical Federated Learning with Central Features ($aggVFL_c$), similar to aggVFL but incorporating its own features; Split Vertical Federated Learning ($splitVFL$) [17, 29, 41], featuring a trainable server model processes intermediate results from passive parties; and Split Vertical Federated Learning without Local Features ($splitVFL_c$), where the server party doesn't provide any features to the model but relies solely on intermediate results from client parties.

Because VFL is a collaboration system that requires parties to exchange gradient or model level information, it has been of great research interest to study communication efficiency, and data privacy protection. Various strategies are adopted to heighten communication efficiency, typically involving reducing the cost of coordination and compressing the data transmitted between parties, such as multiple client updates [78], asynchronous coordination [37], one-shot communication [71], and data compression [5, 37]. In terms of data privacy protection, VFL relies on cutting-edge technologies like Homomorphic Encryption (HE) [74], Multi-Party Computation (MPC) [72, 43], and Differential Privacy (DP) [62] to preserve data privacy.

## A.2  Vertical Federated Adversarial Learning

Emerging research has investigated the distinct challenges posed by adversarial attacks in the context of VFL [24]. Due to the distributed nature, VFL struggles to ensure client trustworthiness and thus renders it highly susceptible to adversarial perturbations, underscoring the pressing need for enhanced VFL model robustness[24], this is particularly evident in neural network models. Prior works have proposed that adversaries (third-party or client party) can generate adversarial samples by introducing manipulated perturbations to raw data or embeddings in the corrupted clients, aiming to mislead the inference of VFL models [48, 69, 53, 17]. However, existing VFL defense mechanisms based on cryptographic [45] and non-cryptographic [41] only concentrate on mitigating inference attacks and backdoor attacks while neglecting adversarial attacks.

## A.3  Threat Model

In the context of VFL, we focus on untargeted adversarial attacks, constructed during the inference phase. The adversary's objective is to corrupt samples whose original prediction is $y_u$, causing the server model to output $\hat{y} \neq y_u$. We categorize these adversarial attacks into two primary scenarios:

- **Malicious (colluding) clients.** In this scenario, we consider the presence of malicious (colluding) clients acting as adversary. During the attack, all malicious clients (one or more) collaboratively and simultaneously generate the adversarial feature partition. The attacks are further classified based on the level of knowledge these clients possess:

  - *White-box adversarial attack.* Under relaxed protocol, clients have access to the server model $f_s$ and the output of all clients $x_{t_c}$. This protocol could occur when the client needs to make interpretable decisions based on the server model's parameters [48, 64, 47]. This implies the malicious clients have the necessary information to calculate the partial gradient to the features.

  - *Black-box adversarial attack.* Under basic VFL protocol, all participants keep their private data (e.g., labels and features), as well as the server model $f_s$ and client models $\{f_{(c)}\}_{c=1}^{C}$ local during inference. Clients can only receive the final prediction results $\hat{y}$ and cannot directly obtain the gradient, thus necessitating the use of black-box methods to approximate it.

- **Third party adversary.** We also consider an adversary as a third party in VFL inference, who can access, replay, and manipulate messages on the communication channel between two endpoints, where embeddings and predictions are exchanged. Third-party adversaries usually cannot achieve access to top model parameters, thus this scenario generally falls under the black-box attack category. Due to resource constraints, previous work assumed that the adversary can corrupt at most $C_a \leq C$ clients [16].

## A.4 Adversarial Training as a Dynamical System

With the link between optimal control and deep learning [39], research recast neural networks as dynamical systems and formulated the robust optimization problem as an optimal control problem [54]:

$$\min_{\Theta^1,\ldots,\Theta_T} \max_{\eta_1,\ldots,\eta_S} \quad \sum_{i=1}^{S} \mathcal{L}(x_{T,i}, y_i) + \sum_{i=1}^{S} \sum_{t=0}^{T-1} R_t(x_{t,i}, \Theta_t)$$

$$\text{subject to} \quad x_{t+1,i} = f^t(x_{t,i}, \Theta_t), \quad i = 1, \ldots, \mathcal{S}, \quad t = 1, \ldots, T-1$$

$$x_{1,i} = f_0(x_{0,i} + \eta_i, \Theta_0), \quad i = 1, \ldots, \mathcal{S}$$

(A.1)

where $x_t \in \mathbb{R}^{d_t}$ represents the states (i.e., the input of the $t$-th layer), $f^t : \mathbb{R}^{d_t} \times \Theta^t \rightarrow \mathbb{R}^{d_{t+1}}$ is the state transition map, $\Theta^t$ are the trainable control parameters, $\Theta$ denotes the concatenation of $(\Theta^t)_{0 \leq t \leq T-1}$, and the initial conditions are provided by the inputs to the network, $x_{0,i}$. According to the two-player Pontryagin Maximum principle, proved in [76], we define the Hamiltonians: $H_0(x, p, \theta, \eta) := p^T f_0(x + \eta, \theta) - R_0(x, \theta)$ and $H_t(x, p, \theta) := p^T f_t(x, \theta) - R_t(x, \theta)$, then there exists an optimal costate trajectory $p_t^*$, satisfied:

$$x_{t+1}^* = \nabla_p \mathcal{H}_t(x_t^*, p_{t+1}^*, \theta_t^*) \quad x_0^* = x_0 + \eta^*$$

(A.2)

$$p_t^* = \nabla_x \mathcal{H}_t(x_t^*, p_{t+1}^*, \theta^{t,*}) \quad p_T^* = -\nabla \mathcal{L}(x_T^*, y)$$

(A.3)

where $\Theta^* := \{\theta^{0,*}, \ldots \theta^{T-1,*}\}$ is the solution of the problem (A.1).

Due to the compositional structure, feed-forward deep neural networks can be viewed as dynamical systems. This approach has been recently explored in several papers, which leverage this interpretation to propose new training algorithms [68, 38, 67, 76].

According to equation A.1, the two-player Pontryagin Maximum principle, proved in [76], gives necessary conditions for an optimal setting of the parameters $\theta^*$, perturbations $\eta_1^*, \ldots, \eta_S^*$, and corresponding trajectories $\{x_{t,i}^*\}$. Define the Hamiltonians

$$H_t(x, p, \theta) := p^\top f_t(x, \theta) - R_t(x, \theta), \quad t = 1, \ldots, T-1$$

$$H_0(x, p, \theta, \eta) := p^\top f_0(x + \eta, \theta) - R_0(x, \theta)$$

(A.4)

The two-player maximum principle says in this case that if $\Phi$, $f_t$, and $R_t$ are twice continuously differentiable, with respect to $x$, uniformly bounded in $x$ and $t$ along with their partial derivatives, and the image sets $\{f_t(x, \theta) | \theta \in \mathbb{R}^{m_t}\}$ and $\{R_t(x, \theta) | \theta \in \mathbb{R}^{m_t}\}$ are convex for all $x$ and $t$, then there exists an optimal costate trajectory $p_t^*$ such that the following dynamics are satisfied

$$x_{t+1,i}^* = \nabla_p H_t(x_{t,i}^*, p_{t+1,i}^*, \theta_t^*), \quad x_{1,i}^* = \nabla_p H_0(x_{0,i}, p_{1,i}^*, \theta_0^*, \eta_i^*)$$

$$p_{t,i}^* = \nabla_x H_t(x_{t,i}^*, p_{t+1,i}^*, \theta_t^*), \quad p_{T,i}^* = -\nabla_x \Phi(x_{T,i}^*, y_i)$$

(A.5)

and the following Hamiltonian condition for all $\theta_t \in \mathbb{R}^{m_t}$ and $\eta_i \in X$

$$H_t(x_{t,i}^*, p_{t+1,i}^*, \theta_t) \leq \sum_{i=1}^{S} H_t(x_{t,i}^*, p_{t+1,i}^*, \theta_t^*), \quad t = 1, \ldots, T-1$$

$$\sum_{i=1}^{S} H_0(x_{t,i}^*, p_{t+1,i}^*, \theta_t, \eta_i^*) \leq \sum_{i=1}^{S} H_0(x_{t,i}^*, p_{t+1,i}^*, \theta_t^*, \eta_i^*) \leq \sum_{i=1}^{S} H_0(x_{t,i}^*, p_{t+1,i}^*, \theta_t^*, \eta_i)$$

(A.6)

These necessary optimality conditions can be used to design an iterative algorithm of the following form. For each data point $i \in \{1, \ldots, S\}$,

1. Compute the state and costate trajectories $\{x_{i,t}\}$ and $\{p_{i,t}\}$ from the dynamics, keeping $\theta_t$ and $\eta_i$ fixed:

$$x_{t+1,i}^{(\eta)} = \nabla_p H_t(x_{t,i}^{(\eta)}, p_{t+1,i}^{(\eta)}, \theta_t)$$

$$x_{1,i}^{(\eta)} = \nabla_p H_0(x_{0,i}, p_{1,i}^{(\eta)}, \theta_0, \eta)$$

2. $p_{t,i}^{(\eta)} = \nabla_x H_t(x_{t,i}^{(\eta)}, p_{t+1,i}^{(\eta)}, \theta_t), p_{T,i}^{(\eta)} = -\nabla_x \Phi(x_{T,i}^{(\eta)}, y_i)$

3. Minimize the Hamiltonian $H_0(x_t, i, pt + 1, i, \theta_t, \eta_i)$ with respect to $\eta_i$

4. Maximize the sum of Hamiltonians $\sum_{i=1}^{S} H_t(x_t, i, pt + 1, i, \theta_t)$ with respect to $\theta_t$ for all $t$

As was noticed as early as [35], it can be seen from the chain rule that the backward costate dynamics are equivalent to backpropagation through the network. With this interpretation, the gradient of the total loss for the $i$-th data point with respect to the adversary $\eta_i$ can be written as $\nabla_\eta f_0(x_{0,i} + \eta_i, \theta_0)^\top p_{1,i}^{(\eta)}$. For a fixed value of $\theta_0$, performing gradient descent on $H_0$ to find a worst-case adversarial perturbation can be expressed as the following updates, where $\alpha > 0$ is a step size:

$$\eta_i^{(\ell+1)} = \eta_i^{(\ell)} - \alpha \nabla_\eta f_0(x_{0,i} + \eta_i^{(\ell)}, \theta_0)^\top p_{1,i}^{(\eta)} \tag{A.7}$$

An important observation made in [76] is that the adversary is only present in the first layer Hamiltonian condition and this function can be minimized by computing gradients only with respect to the first layer of the network. More explicitly, instead of using $p_{\ell,1}^{(\eta)}$, as in the updates above, we could instead use $p_{0,1}^{(\eta)}$ and the updates

$$\eta_i^{(\ell+1)} = \eta_i^{(\ell)} - \alpha \nabla_\eta f_0(x_{0,i} + \eta_i^{(\ell)}, \theta_0)^\top p_{0,1}^{(\eta)} \tag{A.8}$$

This removes the need to do a full backpropagation to recompute the costate $p_{\ell,1}^{(\eta)}$ for every update of $\eta_i^{(\ell)}$, at the cost of now being an approximate gradient.

## A.5   Zeroth Order Optimization

ZOO methods [23, 22] have been developed to effectively solve many ML problems for which obtaining explicit gradient expressions is difficult or infeasible. Such problems include structure prediction tasks, where explicit gradients are challenging to derive [57], as well as bandit and black-box learning problems [56, 42], where obtaining explicit gradients is not feasible. Specifically, ZOO relies solely on function values for optimization, eschewing the need for explicit gradients.

Formally, given a function $f(x)$ with input $x$, the gradient $\nabla f(x)$ can be estimated using ZOO. One common approach is to sample random perturbations $u$ within the domain of $f$ and evaluate the function shifts. The ZO gradient estimator $\hat{\nabla} f(x)$ is given by:

$$\hat{\nabla} f(x) = \frac{1}{q} \sum_{j=1}^{q} [f(x + \mu u_j) - f(x)] \frac{u_j}{\mu} \tag{A.9}$$

where $\mu$ serves as a scaling factor for the random perturbation, while $u_j$ represents the $j$-th random perturbation sampled from a distribution $p$ across the domain of $f$. The parameter $q$ denotes the number of random samples employed for estimation. Normalizing the perturbation by $\frac{u_j}{\mu}$ ensures the estimator's unbiasedness. The expectation of the Zeroth Order (ZO) gradient estimator yields an unbiased estimate of the true gradient, expressed as $E[\hat{\nabla} f(x)] = \nabla f(x)$, provided that the samples $u_j$ are drawn from a distribution with a mean of zero.

The application of ZOO to VFL has been discussed, highlighting its specific properties such as model applicability [79], privacy security concerns [42], and considerations regarding communication cost and computational efficiency [62].

## A.6   Communication Compression

Compression is a pivotal technique in VFL that aims to mitigate communication overhead by reducing the volume of data transmitted among participating parties. In the context of neural network-based VFL algorithms, high-dimensional input vectors are inherently mapped onto lower-dimensional representations, which serve a natural compression purpose. However, to further enhance communication efficiency, specialized dimensionality reduction techniques are often integrated. Several VFL frameworks have been proposed to incorporate compression techniques: AVFL [2] leverages PCA to compress the data before transmission, reducing the communication load. CE-VFL [30] employs both PCA and autoencoders to learn latent representations from raw data, which are then used for model training. SecureBoost+ [10] and eHE-SecureBoost [73] encode encrypted gradients into a compact form, minimizing the number of cryptographic operations and the data transmission size. C-VFL [5] introduces an arbitrary compression scheme to VFL, offering a theoretical analysis of how compression parameters impact the overall system efficiency.

Compression techniques play a critical role in VFL by enabling more efficient data transmission without compromising the integrity of the learning process. The selection of an appropriate compression method is contingent upon the specific requirements of the VFL scenario, including the sensitivity of the data, the computational resources available, and the desired balance between communication efficiency and model performance.

# B Convergence analysis

## B.1 Notations

| Notations | Definitions |
|---|---|
| | Neural Network Classifier |
| $\mathcal{S}$ | The number of samples |
| $f$ | Neural network model |
| $\Theta$ | Model Parameters |
| $x_i, y_i$ | Input sample and corresponding label |
| $\mathcal{B}, B$ | The mini-batch $\mathcal{B}$ with size $B$ |
| $\mathbb{E}$ | Expectation |
| $k \in \{1, 2, \ldots, \mathcal{K}\}$ | Iteration index for parameter updating |
| | Vertical Federated Learning |
| $C$ | The number of clients |
| $f_{(1)}, f_{(2)}, \ldots, f_{(C)}$ | Client models |
| $\theta = \{\theta_{(1)}, \theta_{(2)}, \ldots, \theta_{(C)}\}$ | Client model parameters |
| $f_s$ | Server model |
| $\psi$ | Server model parameters |
| $\mathcal{L}$ | Loss function |
| $f = \{f_s, f_{(1)}, f_{(2)}, \ldots, f_{(C)}\}$ | The complete VFL model |
| $\alpha_\psi$ | Learning rate for server model parameters |
| $\alpha_\theta$ | Learning rate for client model parameters |
| | Adversarial Training |
| $\mathcal{A}$ | Adversarial Loss Function |
| $\mathcal{G}_\mathcal{B}(\eta, \psi, \theta)$ | $\frac{1}{B} \sum_{i \in \mathcal{B}} \mathcal{A}_i(\eta_i, \psi_i, \theta_i)$ |
| $\mathcal{R}(\eta, \psi, \theta)$ | $\frac{1}{\mathcal{S}} \sum_{i \in \mathcal{S}} \mathcal{A}_i(\eta_i, \psi_i, \theta_i)$ |
| $\eta_i^*$ | $\mathrm{argmax}_\eta \, \mathcal{A}_i(\eta, \psi, \theta)$ |
| $\eta$ | Adversarial perturbation |
| $\Pi$ | Projection operator |
| $\alpha_\eta$ | Learning rate for adversarial sample |
| $\ell$ | Iteration index for adversarial sample updates |
| $x_{0,i} = \{x_{0,i,(1)}, x_{0,i,(2)}, \ldots, x_{0,i,(C)}\}$ | The sample $i$ with features from all clients |
| $\eta_i = \{\eta_{i,(1)}, \eta_{i,(2)}, \ldots, \eta_{i,(C)}\}$ | The adversarial perturbation for sample $i$ from all clients |
| | Optimal Control Formulation of Deep Learning |
| $\mathcal{H}_t$ | Hamiltonian function for layer $t$ |
| $p_t = \{p_{t,(1)}, p_{t,(2)}, \ldots, p_{t,(C)}\}$ | Costates at layer $t$ |
| $T$ | Number of layers in the neural network |
| $t = 0, 1, \ldots, T-1$ | Layer indices |
| $f^t$ | State transition map for layer $t$ |
| $x_t = \{x_{t,(1)}, x_{t,(2)}, \ldots, x_{t,(C)}\}$ | States at layer $t$ |
| $\Theta^t$ | Trainable parameters for layer $t$ |
| | Decoupled parallel Backpropagation |
| $\mathcal{M}_K$ | The number of divided modules |
| $t_s$ | The number of server model's layers |
| $t_c$ | The number of client model's layers |
| $f = \{f_1, f_2, \ldots, f_{t_c}, \ldots, f_{T-1}\}$ | The VFL model from layer-wise view |
| $\theta = \{\Theta_1, \Theta_2, \ldots, \Theta_{t_c}\}$ | Client model parameters from layer-wise view |
| $x_{t_c}$ | The output of all clients |
| $f_{\check{\theta}_1}, f_{\check{\mathcal{M}}_1}$ | Client model network excluding the first layer/module |
| | Lazy Sequential Backpropagation |
| $M$ | Number of iterations for full propagations |
| $N$ | Number of iterations for propagations in bottom module |
| $R_t$ | Regularizer for layer $t$ |
| $f_{\check{\Theta}_1}, f_{\check{\mathcal{M}}_1}$ | The VFL model excluding the first layer/module |
| $x_{t,i}^{m,n}$ | The state/output of sample $i$ at layer $t$ in $m, n$ iteration |
| $p_{t,i}^{m,n}$ | The costate/gradient of sample $i$ at layer $t$ in $m, n$ iteration |
| | Zoreth Order Gradient Estimation |
| $\mu$ | Smoothing parameter |
| $\mathbf{u}$ | Random vector |
| $q$ | Query budget for gradient estimation |
| $\{\delta_i^j\}_{j=1}^q$ | Loss difference |
| $\hat{\nabla}\mathcal{A}(\eta, \psi, \theta)$ | Estimation Gradient from ZOO |
| | Compressor |
| $\mathcal{C}(\cdot)_b$ | Compressor compressing information to $b$ bits |
| $\nabla\hat{\mathcal{A}}(\eta, \psi, \theta)$ | Compression Gradient |

Table 10: Table of Notations

## B.2 Assumptions

**Assumption 1.** *Lipschitz Gradient: There exists a constant $K > 0$ such that for all $t \in 1, \ldots, t_c, \ldots, T$, the functions $f_t$, $f_c$, $\mathcal{L}$, and $R_t$ are $K$-Lipschitz in $x$, uniformly in $\theta$ and $\psi$. For all each sample $i \in 1, \ldots, \mathcal{S}$, the function $\nabla_\eta \mathcal{A}_i(\eta, \psi, \theta)$, $\nabla_\psi \mathcal{A}_i(\eta, \psi, \theta)$, $\nabla_\theta \mathcal{A}_i(\eta, \psi, \theta)$ satisfy the following Lipschitz conditions:*

$$||\nabla_\eta \mathcal{A}_i(\eta, \psi', \theta) - \nabla_\eta \mathcal{A}_i(\eta, \psi, \theta)|| \leq L_{\eta\psi}||\psi' - \psi||. \tag{B.1}$$

$$||\nabla_\eta \mathcal{A}_i(\eta, \psi, \theta') - \nabla_\eta \mathcal{A}_i(\eta, \psi, \theta)|| \leq L_{\eta\theta}||\theta' - \theta||. \tag{B.2}$$

$$||\nabla_\psi \mathcal{A}_i(\eta', \psi, \theta) - \nabla_\psi \mathcal{A}_i(\eta, \psi, \theta)|| \leq L_{\psi\eta}||\eta' - \eta||. \tag{B.3}$$

$$||\nabla_\psi \mathcal{A}_i(\eta, \psi, \theta') - \nabla_\psi \mathcal{A}_i(\eta, \psi, \theta)|| \leq L_{\psi\theta}||\theta' - \theta||. \tag{B.4}$$

$$||\nabla_\theta \mathcal{A}_i(\eta', \psi, \theta) - \nabla_\theta \mathcal{A}_i(\eta, \psi, \theta)|| \leq L_{\theta\eta}||\eta' - \eta||. \tag{B.5}$$

$$||\nabla_\theta \mathcal{A}_i(\eta, \psi', \theta) - \nabla_\theta \mathcal{A}_i(\eta, \psi, \theta)|| \leq L_{\theta\psi}||\psi' - \psi||. \tag{B.6}$$

**Assumption 2.** *Unbiased Gradient and Bounded Variance: There exists $\sigma_\psi > 0$ and $\sigma_\theta > 0$, the stochastic gradients are unbiased, i.e. $\mathbb{E}_i \nabla_\psi \mathcal{G}_i(\eta, \psi, \theta) = \nabla_\psi \mathcal{R}(\eta, \psi, \theta), \mathbb{E}_i \nabla_\theta \mathcal{G}_i(\eta, \psi, \theta) = \nabla_\theta \mathcal{R}(\eta, \psi, \theta), i = 1, \ldots, B$ and satisfy:*

$$\mathbb{E}||\nabla_\psi \mathcal{G}_\mathcal{B}(\eta, \psi, \theta) - \nabla_\psi \mathcal{R}(\eta, \psi, \theta)||^2 \leq \sigma_\psi^2. \tag{B.7}$$

$$\mathbb{E}||\nabla_\theta \mathcal{G}_\mathcal{B}(\eta, \psi, \theta) - \nabla_\theta \mathcal{R}(\eta, \psi, \theta)||^2 \leq \sigma_\theta^2. \tag{B.8}$$

Assumption 1, 2 are the basic assumptions for solving the non-convex optimization problem with stochastic gradient descent[63][21].

**Assumption 3.** *Bounded Hessian: The Hessian for $\mathcal{A}_i(\eta, \psi, \theta)$ is bounded, i.e.there exist positive constants $H_\psi$ and $H_\theta$ for $\mathcal{A}_i(\eta, \psi, \theta)$, $\psi$ and $\theta$, the following inequalities holds:*

$$||\nabla_\psi^2 \mathcal{A}_i(\eta_i, \psi, \theta)|| \leq H_\psi. \tag{B.9}$$

$$||\nabla_{[\theta, x_{0,i}]}^2 \mathcal{A}_i(\eta_i, \psi, \theta)|| \leq H_\theta. \tag{B.10}$$

**Assumption 4.** *Bounded Block-coordinate Gradient: The gradient of all the participants' local output w.r.t. their local input is bounded, i.e. for, all $i \in 1, \ldots, \mathcal{S}$ there exist positive constants $Q_\psi$ and $Q_\theta$ satisfies the following inequalities:*

$$||\nabla_{[\psi]} \mathcal{A}_i(\eta_i, \psi, \theta)|| \leq Q_\psi. \tag{B.11}$$

$$||\nabla_\theta \mathcal{A}_i(\eta_i, \psi, \theta)|| \leq Q_\theta. \tag{B.12}$$

Assumption 3, 4 are the fundamental assumptions for bounding the compression loss. Compression introduces errors into the input of the loss function; therefore, with a bounded Hessian, we can determine the maximum effect of these errors on the loss. Additionally, bounding the block-coordinated gradient is a common practice in VFL analysis. This approach helps constrain the entire model's gradient when the gradients of other parts have been bounded [62][5].

**Assumption 5.** *z-Strongly Concave: If function $\mathcal{A}_i(\eta, \psi, \theta)$ is z-strongly concave for $\eta$, then for all $\psi$ and $\theta$, the following inequalities satisfy:*

$$||\eta' - \eta|| \leq (1/z)||\nabla_\eta \mathcal{A}_i(\eta, \psi, \theta)||. \tag{B.13}$$

Assumption 5 made in previous results on convergence of robust training [65] and is justified through the reformulation of robust training as distributionally robust optimization. It helps us to bound the delayed gradient of $\eta$.

## B.3 Proposition

**Proposition 1.** *Under **Assumption 1** and **Assumption 5**, the loss function $\mathcal{R}(\eta', \psi, \theta)$ is $L_\psi$-smooth for $\psi$, $L_\theta$-smooth for $\theta$, and the following inequality holds for all $\psi$, $\psi'$, $\theta$, and $\theta'$:*

$$\mathcal{R}\left(\eta', \psi', \theta'\right) - \mathcal{R}\left(\eta, \psi, \theta\right) \leq \left\langle \nabla_\theta \mathcal{R}\left(\eta, \psi, \theta\right), \theta' - \theta \right\rangle + \frac{L_\theta}{2}\left\|\theta' - \theta\right\|^2$$

$$+ \left\langle \nabla_\psi \mathcal{R}\left(\eta, \psi, \theta\right), \psi' - \psi \right\rangle + \frac{L_\psi}{2}\left\|\psi' - \psi\right\|^2. \tag{B.14}$$

*where $L_\psi = L_{\psi\psi} + \frac{L_{\psi\eta}L_{\eta\psi}}{z}$ and $L_\theta = L_{\theta\theta} + \frac{L_{\theta\eta}L_{\eta\theta}}{z}$. This assumption is consistent with **Proposition 1** in [54]. This can help us to connect the N-loop and M-loop.*

**Proposition 2.** *The classical back-propagation-based gradient descent algorithm can be viewed as an algorithm attempting to solve the PMP[76]. The costate processes $p_t^*$ and the gradient $\nabla_{x_t} \mathcal{A}(\eta, \psi, \theta)$ satisfy the following equation:*

$$p_t = -\nabla_{x_t} \mathcal{A}(\eta, \psi, \theta). \tag{B.15}$$

## B.4 Definition

**Definition 1.** *Compression Error (forward message) Considering sample $i$, we can define the compression error of $\mathcal{C}(\cdot)_b$: $e_{c,i}$, $c \in 1, 2, ..., C$, i.e. $e_{c,i} = \mathcal{C}(x_{t_c,c,i})_b - x_{t_c,c,i}$. We denote the expected norm of the error from the client $c$ at global iteration $k$ as $\mathcal{E}_{c,i}^k = \mathbb{E}\|e_{c,i}^k\|^2$, and $\mathcal{E}^k = \max_c \mathcal{E}_{c,i}^k$. Since all client operations are synchronized, the error from all clients is $e_i^k = (e_{1,i}^k, e_{2,i}^k, ..., e_{C,i}^k)$. Then, the expected norm of the error from all clients:*

$$\mathbb{E}\|e_i^k\|^2 = \mathbb{E}\|(e_{1,i}^k, e_{2,i}^k, ..., e_{C,i}^k)\|^2$$
$$\leq \sum_{c=1}^{C} \mathbb{E}\|e_{c,i}^k\|^2$$
$$\leq C\mathcal{E}^k. \tag{B.16}$$

## B.5 Lemma

**Lemma 3.** *Zeroth-Order Optimization. For arbitrary $f$ in problem $(P)$, the following conditions hold:*
*1) $f_\mu(x)$ is continuously differentiable, its gradient is $L_\mu$-Lipschitz continuous with $L_\mu \leq L$:*

$$\nabla f_\mu(x) = \mathbb{E}_{\mathbf{u}}[\hat{\nabla} f(x)], \tag{B.17}$$

*where $\mathbf{u}$ is drawn from the uniform distribution over the unit Euclidean sphere, $f_\mu(x) = \mathbb{E}(f(x + \mu\mathbf{u}))$ is the smooth approximation of $f$.*
*2) For any $x \in \mathbb{R}^d$, the following inequalities satisfy:*

$$\|\nabla f_\mu(x) - \nabla f(x)\|^2 \leq \frac{L^2\mu^2 d^2}{4}. \tag{B.18}$$

Proof of this lemma is provided in [42, 18].

**Lemma 4.** *Bound the costate difference at the bottom module. There exists a constant $G'$ dependent on $T$ and $K$ such that for all $n \in \{0, \ldots, N\}$, $m \in \{0, \ldots, M\}$, and $i \in \{1, \ldots, S\}$:*

$$\left\|p_{\mathcal{M}_1,i}^{m-\tau_1,0} - p_{\mathcal{M}_1,i}^{m,N}\right\| \leq G'\alpha_\eta \left(\mathcal{M}_K N - 1\right). \tag{B.19}$$

*Where $G' = TK^{T+1}(K^T + T(T-1)K^{2T-2} + TK^{2T})$, $m$ is the iteration index of full propagation.*

**Proof:** This lemma bounds the difference of the costates of the first module in the adversary's $N$-loop. We fix the data point $i$, and for ease of notation drop the dependence of state variables on the index $i$, while also suppressing the notational dependence on $\Theta$ for all functions, as $\Theta$ is fixed during the updates for the adversary $\eta$. We define $x_t$ and $p_t$ as the state and costate trajectories generated from the initial condition $x_0 + \eta$. We additionally define $\delta p_t^\ell := p_t^0 - p_t^\ell$ and $\delta x_t^\ell := x_t^0 - x_t^\ell$, $\ell$ is the iteration index of example updates. We first prove bounds on $\|p_t^\ell\|$ and $\|\delta x_t^\ell\|$.

Applying Assumption (1), we have:

$$\|p_T^\ell\| \leq \|-\nabla\Phi(x_T^\ell, y)\| \leq K. \tag{B.20}$$
$$\|p_t^\ell\| = \|\nabla_x \mathcal{H}_t(x_t^\ell, p_{t+1}^\ell, \theta_t)\|$$
$$\leq \|p_{t+1}^\ell\|\|\nabla_x f_t(x_t^\ell, \theta_t)\| + \|\nabla_x R_t(x_t^\ell)\|$$
$$\leq K\|p_{t+1}^\ell\| + K$$
$$\leq K + K^2 + \ldots + K^{T-t+1}$$
$$\leq K^{T-t+1}(T - t + 1). \tag{B.21}$$

Next, from Assumption (1), we have $\|\delta x_1^\ell\| = \|f_1(x_0 + \eta^0) - f_1(x_0 + \eta^\ell)\| \leq K\|\eta^0 - \eta^\ell\|$. By induction, we have:

$$\|\delta x_t^\ell\| \leq K^t\|\eta^0 - \eta^\ell\|. \tag{B.22}$$

To bound $\|p_{\mathcal{M}_1}^0 - p_{\mathcal{M}_1}^\ell\|$, we first note that $\|\delta p_T^\ell\| = \|\nabla\Phi(x_T^\ell) - \nabla\Phi(x_T^0)\| \leq K\|\delta x_T^\ell\|$. We write:

$$\|\delta p_t^\ell\| = \|\nabla_x H_t(x_t^0, p_{t+1}^0) - \nabla_x H_t(x_t^\ell, p_{t+1}^\ell)\|$$

$$= \|\nabla_x H_t(x_t^0, p_{t+1}^0) - \nabla_x H_t(x_t^\ell, p_{t+1}^0) + \nabla_x H_t(x_t^\ell, p_{t+1}^0) - \nabla_x H_t(x_t^\ell, p_{t+1}^\ell)\|$$

$$= \|\langle p_{t+1}^0, \nabla_x f_t(x_t^0) - \nabla_x f_t(x_t^\ell)\rangle + \langle p_{t+1}^0 - p_{t+1}^\ell, \nabla_x f_t(x_t^\ell)\rangle + \nabla_x R_t(x_t^\ell) - \nabla_x R_t(x_t^0)\|$$

$$\leq K^{T-1}\left(K\|\delta x_T^\ell\| + \sum_{t=1}^{T-1}(K^{T-t+1}(T-t)+K)\|\delta x_t^\ell\|\right). \tag{B.23}$$

Applying (B.22), we have:

$$\|\delta p_{\mathcal{M}_1}^\ell\| \leq (K^T + T(T-1)K^{2T-2} + TK^{2T})\|\eta^0 - \eta^\ell\|. \tag{B.24}$$

$\eta$ updates with the form:

$$\eta^{\ell+1} = \eta^\ell - \alpha_\eta \nabla_\eta f_{\mathcal{M}_1}(x_0 + \eta^\ell, \theta_{\mathcal{M}_1})^\top p_{\mathcal{M}_1}^0. \tag{B.25}$$

Applying Assumption (1) and (B.21), we have:

$$\|\eta^0 - \eta^\ell\| \leq K^{T+1}T\alpha_\eta(\ell-1). \tag{B.26}$$

Finally, substituting with (B.26) gives us the desired result:

$$\|p_{\mathcal{M}_1,i}^0 - p_{\mathcal{M}_1,i}^\ell\| \leq G'\alpha_\eta(\ell-1), \tag{B.27}$$

where $G' = TK^{T+1}(K^T + T(T-1)K^{2T-2} + TK^{2T})$.

Then, We are going to bound $\left\|p_{\mathcal{M}_1,i}^{m-\tau_1,0} - p_{\mathcal{M}_1,i}^{m,N}\right\|$:

$$\left\|p_{\mathcal{M}_1,i}^{m-\tau_1,0} - p_{\mathcal{M}_1,i}^{m,N}\right\| = \left\|p_{\mathcal{M}_1,i}^{m-\tau_1,0} - p_{\mathcal{M}_1,i}^{m,0} + p_{\mathcal{M}_1,i}^{m,0} - p_{\mathcal{M}_1,i}^{m,N}\right\|$$

$$\overset{(a)}{\leq} \left\|p_{\mathcal{M}_1,i}^{m-\tau_1,0} - p_{\mathcal{M}_1,i}^{m,0}\right\| + \left\|p_{\mathcal{M}_1,i}^{m,0} - p_{\mathcal{M}_1,i}^{m,N}\right\|$$

$$\overset{(b)}{\leq} G'\alpha_\eta(\tau_1 N) + G'\alpha_\eta(N-1)$$

$$\leq G'\alpha_\eta\left[(\tau_1+1)N - 1\right]$$

$$\leq G'\alpha_\eta\left[\mathcal{M}_K N - 1\right], \tag{B.28}$$

where, $(a)$ is obtained using the triangle inequality, $(b)$ is obtained using (B.27), for each $M$-loop, the adversary is updated $N$ times. Proof completes.

**Lemma 5.** ***Bound Compression Error.*** *Under **Assumption 3, 4**, and **Definition 1**, the norm of the difference between the loss function value with and without compression error is bounded:*

$$\mathbb{E}\|\nabla_\psi \hat{\mathcal{A}}_i(\eta,\psi,\theta) - \nabla_\psi \mathcal{A}_i(\eta,\psi,\theta)\| \leq H_\psi^2 C\mathcal{E}^k, \tag{B.29}$$

$$\mathbb{E}\|\nabla_\theta \hat{\mathcal{A}}_i(\eta,\psi,\theta) - \nabla_\theta \mathcal{A}_i(\eta,\psi,\theta)\| \leq Q_\theta^2 H_\theta^2 C\mathcal{E}^k, \tag{B.30}$$

$$\mathbb{E}\|\nabla_{x_{t_c}} \hat{\mathcal{A}}_i(\eta,\psi,\theta) - \nabla_{x_{t_c}} \mathcal{A}_i(\eta,\psi,\theta)\| \leq H_\theta^2 C\mathcal{E}^k. \tag{B.31}$$

The proof of this lemma proceeds same to **Lemma D.4** in [62].

**Lemma 6.** ***Bound the gradient for*** $\eta$. *Due to the communication between the clients and the server involved in the update process of adversarial examples, the gradient $\nabla_\eta \mathcal{A}$ respect to $\eta$ involves estimation gradient $\nabla_\eta \hat{\mathcal{A}}$ from ZOO and compression gradient $\hat{\nabla}_\eta \mathcal{A}$. Under the **Assumption 1**, and **Lemma 3, 5**, at the $i$-th sample and $k$-th iteration, the pseudo-partial derivative for $\eta$ satisfies the following inequality:*

$$\hat{\eta}_i = \underset{\substack{m=1,\ldots,M \\ n=1,\ldots,N}}{argmin}\left\|\hat{\nabla}_\eta \hat{\mathcal{A}}_i\left(\eta_i^{m,n},\psi_i,\theta_i\right)\right\|,$$

$$\mathbb{E}\left\|\hat{\nabla}_\eta \hat{\mathcal{A}}_i(\hat{\eta}_i,\psi_i,\theta_i)\right\|^2 \leq \left[D(\mathcal{X})L_{\eta\eta}^2\left(1-\frac{z}{L_{\eta\eta}}\right)^{MN+1} + \frac{2G^2}{L_{\eta\eta}}[(\mathcal{M}_K N-1]^2\left(\frac{2}{z}+\frac{1}{2L_{\eta\eta}}\right)\right]$$

$$\times 3\left(H_\theta^2 C\mathcal{E}^k + \frac{L^2\mu^2 d^2}{4} + K^2\right). \tag{B.32}$$

*where $G = KG'$, $\alpha_x < \frac{1}{L_{\eta\eta}}$, and $\eta \in \mathcal{X}$.*

**Proof:**

According to the chain rule, we note that $\hat{\nabla}_\eta \hat{\mathcal{A}}_i(\hat{\eta}_i, \psi_i, \theta_i)$ can be split as follows:

$$\mathbb{E}\left\|\hat{\nabla}_\eta \hat{\mathcal{A}}_i(\hat{\eta}_i, \psi_i, \theta_i)\right\|^2 = \mathbb{E}||\nabla_\eta x_{t_c,i} \hat{\nabla}_{x_{t_c}} \hat{\mathcal{A}}_i(\hat{\eta}, \psi_i, \theta_i)||^2$$

$$\leq \underbrace{\mathbb{E}||\nabla_\eta x_{t_c,i}||^2}_{a} \underbrace{\mathbb{E}||\hat{\nabla}_{x_{t_c}} \hat{\mathcal{A}}_i(\hat{\eta}, \psi_i, \theta_i)||^2}_{b}. \tag{B.33}$$

For (a): we view the clients' networks as an independent model. From **Proposition 2**, we can get the following:

$$||p_{t_c,i}^{m,n}|| = ||-\nabla_{x_{t_c}} \mathcal{A}_i(\eta_i^{m,n}, \psi_i, \theta_i)|| \leq K, \tag{B.34}$$

where $m = 1, 2, ..., M$ denotes $M$-loop index, $n = 1, 2, ..., N$ denotes $N$-loop index.

According to the **Lemma 8** in [54], we drop the dependence of all functions on $\Theta$ and the data point index $i$ for the proof. The $N$-loop of the adversary's updates can be written as (B.25). Recall that the true gradient of $\mathcal{A}(\eta^{m,N})$ is

$$\nabla_\eta \mathcal{A}(\eta^{m,N}) = \nabla_\eta f_{\mathcal{M}_1}(x + \eta)^\top p_{\mathcal{M}_1}^{m,N}. \tag{B.35}$$

We will bound the maximum difference of the update vector to the true gradient over the iterations of the adversary's updates. In this sense, the adversary's updates can be viewed as a standard gradient method with an inexact gradient oracle. We write

$$||\nabla_\eta f_{\mathcal{M}_1}(x + \eta)^\top p_{\mathcal{M}_1}^{m-\tau,0} - \nabla_\eta \mathcal{A}(\eta^{m,N})|| = ||\nabla_\eta f_{\mathcal{M}_1}(x + \eta)^\top p_{\mathcal{M}_1}^{m-\tau,0} - \nabla_\eta f_{\mathcal{M}_1}(x + \eta)^\top p_{\mathcal{M}_1}^{m,N}||$$

$$\leq ||p_1^{m-\tau,0} - p_1^{m,N}|| ||\nabla_\eta f_{\mathcal{M}_1}(x + \eta)^\top||$$

$$\leq KG'\alpha_\eta \left[\mathcal{M}_K N - 1\right]$$

$$= G\alpha_\eta \left[\mathcal{M}_K N - 1\right]. \tag{B.36}$$

We now appeal to an inexact oracle convergence result in [15]. Given a concave function $f(x')$ and a point $x'$, we define a $(\delta, \mu, L)$ oracle as returning a vector $g(x')$ such that the following inequality holds:

$$\frac{\mu}{2}||x' - x||^2 \leq f(x') - f(x) + \langle g(x'), x' - x \rangle \leq \frac{L}{2}||x' - x||^2 + \delta. \tag{B.37}$$

It can be shown that if we have an approximate gradient bound of the form (B.36), and $\mathcal{A}$ is $L_{\eta\eta}$-smooth (Assumption 1) and $z$-strongly concave in $\eta$ (Assumption 5), then the updates for the adversary are created by a $(\delta, z/2, 2L_{\eta\eta})$-oracle, where

$$\delta = G^2 \alpha_\eta^2 \left[\mathcal{M}_K N - 1\right]^2 \left(\frac{2}{z} + \frac{1}{2L_{\eta\eta}}\right). \tag{B.38}$$

Letting $\alpha_\eta < 1/L_{\eta\eta}$ and applying Theorem 4 in [15], along with the inequality $||\nabla A(\hat{\eta})||^2 \leq 2L_{\eta\eta}(\max_\eta A(\eta) - A(\hat{\eta}))$ from the $L_{\eta\eta}$ smoothness of $A$ in $\eta$ gives

$$||\nabla_\eta A(\hat{\eta}, \theta)||^2 \leq L_{\eta\eta}^2 ||\eta^{0,0} - \eta^*||^2 \left(1 - \frac{z}{L_{\eta\eta}}\right)^{MN+1} + \frac{2G^2}{L_{\eta\eta}} \left[\mathcal{M}_K N - 1\right]^2 \left(\frac{2}{z} + \frac{1}{2L_{\eta\eta}}\right)$$

$$\leq D(\mathcal{X}) L_{\eta\eta}^2 \left(1 - \frac{z}{L_{\eta\eta}}\right)^{MN+1} + \frac{2G^2}{L_{\eta\eta}} \left[\mathcal{M}_K N - 1\right]^2 \left(\frac{2}{z} + \frac{1}{2L_{\eta\eta}}\right), \tag{B.39}$$

where $\eta^*$ is the true solution to the inner maximization problem. Since we initialize $\eta^{0,0} \in \mathcal{X}$, we have that $||\eta^{0,0} - \eta^*||^2 \leq D(\mathcal{X})$. We can get:

$$\mathbb{E}||\nabla_\eta x_{t_c,i}||^2 \leq D(\mathcal{X}) L_{\eta\eta}^2 \left(1 - \frac{z}{L_{\eta\eta}}\right)^{MN+1} + \frac{2G^2}{L_{\eta\eta}} \left[\mathcal{M}_K N - 1\right]^2 \left(\frac{2}{z} + \frac{1}{2L_{\eta\eta}}\right). \tag{B.40}$$

For (b): we use **Lemma 3**, and **Assumption 1**:

$$\mathbb{E}||\hat{\nabla}_{x_{t_c}} \hat{\mathcal{A}}_i(\hat{\eta}_i, \psi_i, \theta_i)||^2$$

$$\leq 3\mathbb{E}||\hat{\nabla}_{x_{t_c}} \hat{\mathcal{A}}_i(\hat{\eta}_i, \psi_i, \theta_i) - \hat{\nabla}_{x_{t_c}} \mathcal{A}_i(\hat{\eta}_i, \psi_i, \theta_i)||^2 + 3\mathbb{E}||\hat{\nabla}_{x_{t_c}} \mathcal{A}_i(\hat{\eta}_i, \psi_i, \theta_i) - \nabla_{x_{t_c}} \mathcal{A}_i(\hat{\eta}_i, \psi_i, \theta_i)||^2$$

$$+ 3\mathbb{E}||\nabla_{x_{t_c}} \mathcal{A}_i(\hat{\eta}_i, \psi_i, \theta_i)||^2$$

$$\leq 3(H_\theta^2 C \mathcal{E}^k + \frac{L^2 \mu^2 d^2}{4} + K^2). \tag{B.41}$$

Substituting (a) and (b) completes the proof.

**Lemma 7.** *Connecting Gradients. Under the **Assumption 1**, **Assumption 5**, and **Lemma 2**, the following inequality can be obtained:*

$$\mathbb{E}||\nabla_\psi \mathcal{G}_\mathcal{B}(\hat{\eta}, \psi, \theta) - \nabla_\psi \mathcal{G}_\mathcal{B}(\eta^*, \psi, \theta)||^2 \leq \frac{L_{\psi\eta}^2 \cdot \zeta^k}{z^2}, \tag{B.42}$$

$$\mathbb{E}||\nabla_\theta \mathcal{G}_\mathcal{B}(\hat{\eta}, \psi, \theta) - \nabla_\theta \mathcal{G}_\mathcal{B}(\eta^*, \psi, \theta)||^2 \leq \frac{L_{\theta\eta}^2 \cdot \zeta^k}{z^2}, \tag{B.43}$$

*where $\zeta^k = 3(H_\theta^2 C \mathcal{E}^k + \frac{L^2 \mu^2 d^2}{4} + K^2)\pi(M, N)$,*
$\pi(M, N) = \left\{ D(\mathcal{X})L_{\eta\eta}^2 \left(1 - \frac{z}{L_{\eta\eta}}\right)^{MN+1} + \frac{2G^2}{L_{\eta\eta}} [\mathcal{M}_K N - 1]^2 \left(\frac{2}{z} + \frac{1}{2L_{\eta\eta}}\right) \right\}.$

**Proof:**

Under **Assumption 1**, **Assumption 5**, and **Lemma 2**, for server model parameters $\psi$, we can get:

$$\mathbb{E}||\nabla_\psi \mathcal{G}_\mathcal{B}(\hat{\eta}, \psi, \theta) - \nabla_\psi \mathcal{G}_\mathcal{B}(\eta^*, \psi, \theta)||^2 \leq \frac{1}{B} \sum_{i \in \mathcal{B}} \mathbb{E}||\nabla_\psi \mathcal{A}_i(\hat{\eta}_i, \psi_i, \theta_i) - \nabla_\psi \mathcal{A}_i(\eta_i^*, \psi_i, \theta_i)||^2$$

$$\leq \frac{L_{\psi\eta}^2}{B} \sum_{i \in \mathcal{B}} \mathbb{E}||\hat{\eta}_i - \eta_i^*||^2$$

$$\leq \frac{L_{\psi\eta}^2}{Bz^2} \sum_{i \in \mathcal{B}} \mathbb{E}||\nabla_\eta \mathcal{A}_i(\hat{\eta}_i, \psi_i, \theta_i)||^2$$

$$\leq \frac{L_{\psi\eta}^2 \cdot \zeta^k}{z^2}. \tag{B.44}$$

Similar to the proof for $\psi$ in (B.44), for client model parameters $\theta$, we get:

$$\mathbb{E}||\nabla_\theta \mathcal{G}_\mathcal{B}(\hat{\eta}, \psi, \theta) - \nabla_\theta \mathcal{G}_\mathcal{B}(\eta^*, \psi, \theta)||^2 \leq \frac{L_{\theta\eta}^2 \cdot \zeta^k}{z^2}. \tag{B.45}$$

## B.6 Theorem

**Theorem 1.** *Bound the Global Update Round. When the parameters are updated with the perturbations:*

$$\hat{\eta}_i = \underset{\substack{m=1,\ldots,M \\ n=1,\ldots,N}}{argmin} \left\| \hat{\nabla}_\eta \hat{\mathcal{A}}_i \left( \eta_i^{m,n}, \psi_i, \theta_i \right) \right\|. \tag{B.46}$$

*The gradient of $\hat{\eta}_i$ is bounded:*

$$\mathbb{E}||\hat{\nabla}_\eta \hat{\mathcal{A}}_i(\hat{\eta}_i, \psi_i, \theta_i)||^2 \leq \zeta^k, \tag{B.47}$$

*where $\zeta^k = 3(H_\theta^2 C \mathcal{E}^k + \frac{L^2 \mu^2 d^2}{4} + K^2)\pi(M, N)$,*
$\pi(M, N) = \left\{ D(\mathcal{X})L_{\eta\eta}^2 \left(1 - \frac{z}{L_{\eta\eta}}\right)^{MN+1} + \frac{2G^2}{L_{\eta\eta}} [\mathcal{M}_K N - 1]^2 \left(\frac{2}{z} + \frac{1}{2L_{\eta\eta}}\right) \right\}.$

*The global iterates satisfy:*

$$\frac{1}{\mathcal{K}} \sum_{k=0}^{\mathcal{K}-1} \mathbb{E}\left[ ||\nabla \mathcal{R}\left(\eta^{*,k}, \psi^k, \theta^k\right)||^2 \right]$$

$$\leq \left(\frac{2\Lambda}{\alpha_m \mathcal{K}}\right) + \left(\frac{2L_\psi \alpha_M^2 \sigma_\psi^2}{\alpha_m} + \frac{2L_\theta \alpha_M^2 \sigma_\theta^2}{\alpha_m}\right)$$

$$+ \mathcal{E}\frac{\alpha_M}{\alpha_m} \left[ 2(1 + L_\psi \alpha_M)H_\psi^2 C + 3(1 + L_\theta \alpha_M)Q_\theta^2 H_\theta^2 C + \frac{3H_\theta^2 C \pi(M, N)}{z^2}(2(1 + L_\psi \alpha_M)L_{\psi\eta}^2 + 3(1 + L_\theta \alpha_M)L_{\theta\eta}^2) \right]$$

$$+ \mu^2 \left(\frac{3\alpha_M(1 + L_\theta \alpha_M)L^2 d^2}{4\alpha_m} + \frac{3\pi(M, N)L^2 d^2 a_M(1 + L_\psi \alpha_M)L_{\psi\eta}^2}{2a_m z^2} + \frac{9\pi(M, N)L^2 d^2 a_M(1 + L_\theta \alpha_M)L_{\theta\eta}^2}{4a_m z^2}\right)$$

$$+ \frac{3\alpha_M K^2 \pi(M, N)}{\alpha_m z^2} \left[2(1 + L_\psi \alpha_M)L_{\psi\eta}^2 + 3(1 + L_\theta \alpha_M)L_{\theta\eta}^2\right]. \tag{B.48}$$

**Proof:**

For the gradient respect to $\psi$, there exists compression error, but no estimation error: $\nabla_\psi \hat{\mathcal{G}}_\mathcal{B}(\hat{\eta}^k, \psi^k, \theta^k) := (1/B) \sum_{i \in \mathcal{B}} \nabla_\psi \hat{\mathcal{A}}_i(\hat{\eta}_i^k, \psi_i^k, \theta_i^k)$, where $\hat{\eta}_i^k$ is the output of the adversary's inner problem at iteration $k$, $\hat{\eta}_i^k$ and

$\nabla_\psi \widehat{\mathcal{G}}_\mathcal{B}(\hat{\eta}^k, \psi^k, \theta^k)$ satisfy the following equations:

$$\hat{\eta}_i = \underset{\substack{m=1,\ldots,M \\ n=1,\ldots,N}}{argmin} \left\| \hat{\nabla}_\eta \hat{\mathcal{A}}_i \left( \eta_i^{m,n}, \psi_i, \theta_i \right) \right\| \tag{B.49}$$

$$\psi^{k+1} = \psi^k - \alpha_\psi \cdot \nabla_\psi \widehat{\mathcal{G}}_\mathcal{B}(\hat{\eta}^k, \psi^k, \theta^k). \tag{B.50}$$

For the gradient respect to $\theta$, there exist compression error and estimation error: $\hat{\nabla}_\theta \hat{\mathcal{G}}_\mathcal{B}(\hat{\eta}^k, \psi^k, \theta^k) := (1/B) \sum_{i \in \mathcal{B}} \hat{\nabla}_\theta \hat{\mathcal{A}}_i(\hat{\eta}_i^k, \psi_i^k, \theta_i^k)$, the $\hat{\nabla}_\theta \hat{\mathcal{G}}_\mathcal{B}(\hat{\eta}^k, \psi^k, \theta^k)$ satisfy the following equation:

$$\theta^{k+1} = \theta^k - \alpha_\theta \cdot \hat{\nabla}_\theta \hat{\mathcal{G}}_\mathcal{B}(\hat{\eta}^k, \psi^k, \theta^k). \tag{B.51}$$

Furthermore, $\nabla_\psi \mathcal{G}_\mathcal{B}(\eta^{*,k}, \psi^k, \theta^k)$ and $\nabla_\theta \mathcal{G}_\mathcal{B}(\eta^{*,k}, \psi^k, \theta^k)$ are true stochastic gradients, $\nabla_\psi \mathcal{R}(\eta^{*,k}, \psi^k, \theta^k)$ and $\nabla_\theta \mathcal{R}(\eta^{*,k}, \psi^k, \theta^k)$ are true full gradients.
We begin with the inequality for the $L$-smoothness of $\nabla \mathcal{R}(\eta^{*,k}, \psi^k, \theta^k)$, and apply **Proposition1**, $k \in 0, 1, \ldots, \mathcal{K}$ is the iteration indice, we can get:

$$\mathcal{R}\left( \eta^{*,k+1}, \psi^{k+1}, \theta^{k+1} \right) - \mathcal{R}\left( \eta^{*,k}, \psi^k, \theta^k \right)$$
$$\leq \underbrace{\left\langle \nabla_\psi \mathcal{R}\left( \eta^{*,k}, \psi^k, \theta^k \right), \psi^{k+1} - \psi^k \right\rangle + \frac{L_\psi}{2} \left\| \psi^{k+1} - \psi^k \right\|^2}_{a}$$
$$+ \underbrace{\left\langle \nabla_\theta \mathcal{R}\left( \eta^{*,k}, \psi^k, \theta^k \right), \theta^{k+1} - \theta^k \right\rangle + \frac{L_\theta}{2} \left\| \theta^{k+1} - \theta^k \right\|^2}_{b}. \tag{B.52}$$

For (a):

$$\left\langle \nabla_\psi \mathcal{R}\left( \eta^{*,k}, \psi^k, \theta^k \right), \psi^{k+1} - \psi^k \right\rangle + \frac{L_\psi}{2} \left\| \psi^{k+1} - \psi^k \right\|^2$$
$$= \left\langle \nabla_\psi \mathcal{R}\left( \eta^{*,k}, \psi^k, \theta^k \right), -\alpha_\psi \cdot \nabla_\psi \widehat{\mathcal{G}}_\mathcal{B}(\hat{\eta}^k, \psi^k, \theta^k) \right\rangle + \frac{L_\psi \alpha_\psi^2}{2} \left\| \nabla_\psi \widehat{\mathcal{G}}_\mathcal{B}(\hat{\eta}^k, \psi^k, \theta^k) \right\|^2$$
$$= -\alpha_\psi(1 - \frac{L_\psi \alpha_\psi}{2})||\nabla_\psi \mathcal{R}\left( \eta^{*,k}, \psi^k, \theta^k \right)||^2 + \frac{L_\psi \alpha_\psi^2}{2}||\nabla_\psi \widehat{\mathcal{G}}_\mathcal{B}(\hat{\eta}^k, \psi^k, \theta^k) - \nabla_\psi \mathcal{R}\left( \eta^{*,k}, \psi^k, \theta^k \right)||^2$$
$$+ \alpha_\psi(1 - L_\psi \alpha_\psi)\left\langle \nabla_\psi \mathcal{R}\left( \eta^{*,k}, \psi^k, \theta^k \right), \nabla_\psi \mathcal{R}\left( \eta^{*,k}, \psi^k, \theta^k \right) - \nabla_\psi \widehat{\mathcal{G}}_\mathcal{B}(\hat{\eta}^k, \psi^k, \theta^k) \right\rangle$$
$$= -\alpha_\psi(1 - \frac{L_\psi \alpha_\psi}{2})||\nabla_\psi \mathcal{R}\left( \eta^{*,k}, \psi^k, \theta^k \right)||^2 + \frac{L_\psi \alpha_\psi^2}{2}||\nabla_\psi \widehat{\mathcal{G}}_\mathcal{B}(\hat{\eta}^k, \psi^k, \theta^k) - \nabla_\psi \mathcal{R}\left( \eta^{*,k}, \psi^k, \theta^k \right)||^2$$
$$+ \alpha_\psi(1 - L_\psi \alpha_\psi)\left\langle \nabla_\psi \mathcal{R}\left( \eta^{*,k}, \psi^k, \theta^k \right), \nabla_\psi \mathcal{R}\left( \eta^{*,k}, \psi^k, \theta^k \right) - \nabla_\psi \mathcal{G}_\mathcal{B}(\eta^{*,k}, \psi^k, \theta^k) \right\rangle$$
$$+ \alpha_\psi(1 - L_\psi \alpha_\psi)\left\langle \nabla_\psi \mathcal{R}\left( \eta^{*,k}, \psi^k, \theta^k \right), \nabla_\psi \mathcal{G}_\mathcal{B}(\eta^{*,k}, \psi^k, \theta^k) - \nabla_\psi \widehat{\mathcal{G}}_\mathcal{B}(\hat{\eta}^k, \psi^k, \theta^k) \right\rangle$$
$$\leq -\alpha_\psi(1 - \frac{L_\psi \alpha_\psi}{2})||\nabla_\psi \mathcal{R}\left( \eta^{*,k}, \psi^k, \theta^k \right)||^2 + L_\psi \alpha_\psi^2 ||\nabla_\psi \widehat{\mathcal{G}}_\mathcal{B}(\hat{\eta}^k, \psi^k, \theta^k) - \nabla_\psi \mathcal{G}_\mathcal{B}(\eta^{*,k}, \psi^k, \theta^k)||^2$$
$$+ L_\psi \alpha_\psi^2 ||\nabla_\psi \mathcal{G}_\mathcal{B}(\eta^{*,k}, \psi^k, \theta^k) - \nabla_\psi \mathcal{R}\left( \eta^{*,k}, \psi^k, \theta^k \right)||^2$$
$$+ \alpha_\psi(1 - L_\psi \alpha_\psi)\left\langle \nabla_\psi \mathcal{R}\left( \eta^{*,k}, \psi^k, \theta^k \right), \nabla_\psi \mathcal{R}\left( \eta^{*,k}, \psi^k, \theta^k \right) - \nabla_\psi \mathcal{G}_\mathcal{B}(\eta^{*,k}, \psi^k, \theta^k) \right\rangle$$
$$+ \frac{\alpha_\psi}{2}(1 - L_\psi \alpha_\psi)||\nabla_\psi \mathcal{R}\left( \eta^{*,k}, \psi^k, \theta^k \right)||^2 + \frac{\alpha_\psi}{2}(1 - L_\psi \alpha_\psi)||\nabla_\psi \mathcal{G}_\mathcal{B}(\eta^{*,k}, \psi^k, \theta^k) - \nabla_\psi \widehat{\mathcal{G}}_\mathcal{B}(\hat{\eta}^k, \psi^k, \theta^k)||^2$$
$$= -\frac{\alpha_\psi}{2}||\nabla_\psi \mathcal{R}\left( \eta^{*,k}, \psi^k, \theta^k \right)||^2 + \frac{\alpha_\psi}{2}(1 + L_\psi \alpha_\psi)||\nabla_\psi \widehat{\mathcal{G}}_\mathcal{B}(\hat{\eta}^k, \psi^k, \theta^k) - \nabla_\psi \mathcal{G}_\mathcal{B}(\eta^{*,k}, \psi^k, \theta^k)||^2$$
$$+ L_\psi \alpha_\psi^2 ||\nabla_\psi \mathcal{G}_\mathcal{B}(\eta^{*,k}, \psi^k, \theta^k) - \nabla_\psi \mathcal{R}\left( \eta^{*,k}, \psi^k, \theta^k \right)||^2$$
$$+ \alpha_\psi(1 - L_\psi \alpha_\psi)\left\langle \nabla_\psi \mathcal{R}\left( \eta^{*,k}, \psi^k, \theta^k \right), \nabla_\psi \mathcal{R}\left( \eta^{*,k}, \psi^k, \theta^k \right) - \nabla_\psi \mathcal{G}_\mathcal{B}(\eta^{*,k}, \psi^k, \theta^k) \right\rangle. \tag{B.53}$$

Note that $\mathbb{E}\left[ \nabla_\psi \mathcal{G}_\mathcal{B}(\eta^{*,k}, \psi^k, \theta^k) \right] = \nabla_\psi \mathcal{R}\left( \eta^{*,k}, \psi^k, \theta^k \right)$, where the expectation is taken over the randomness of the mini-batch sampling. We can get:

$$\mathbb{E}\left[ \nabla_\psi \mathcal{R}\left( \eta^{*,k}, \psi^k, \theta^k \right), \nabla_\psi \mathcal{R}\left( \eta^{*,k}, \psi^k, \theta^k \right) - \nabla_\psi \mathcal{G}_\mathcal{B}(\eta^{*,k}, \psi^k, \theta^k) \right] = 0. \tag{B.54}$$

Then, we can get:

$$\mathbb{E}\left[\left\langle \nabla_\psi \mathcal{R}\left(\eta^{*,k}, \psi^k, \theta^k\right), \psi^{k+1} - \psi^k\right\rangle + \frac{L_\psi}{2}\left\|\psi^{k+1} - \psi^k\right\|^2\right]$$

$$\leq -\frac{\alpha_\psi}{2}\mathbb{E}\left[||\nabla_\psi \mathcal{R}\left(\eta^{*,k}, \psi^k, \theta^k\right)||^2\right] + \frac{\alpha_\psi}{2}(1 + L_\psi\alpha_\psi)\mathbb{E}\left[||\nabla_\psi\widehat{\mathcal{G}}_\mathcal{B}(\hat\eta^k, \psi^k, \theta^k) - \nabla_\psi\mathcal{G}_\mathcal{B}(\eta^{*,k}, \psi^k, \theta^k)||^2\right]$$

$$+ L_\psi\alpha_\psi^2\mathbb{E}\left[||\nabla_\psi\mathcal{G}_\mathcal{B}(\eta^{*,k}, \psi^k, \theta^k) - \nabla_\psi\mathcal{R}\left(\eta^{*,k}, \psi^k, \theta^k\right)||^2\right]. \tag{B.55}$$

Under **Assumption 2**, we can get:

$$\mathbb{E}\left[||\nabla_\psi\mathcal{G}_\mathcal{B}(\eta^{*,k}, \psi^k, \theta^k) - \nabla_\psi\mathcal{R}\left(\eta^{*,k}, \psi^k, \theta^k\right)||^2\right] \leq \sigma_\psi^2. \tag{B.56}$$

Furthermore, under **Lemma 5** and **Lemma 7**, we can get:

$$\mathbb{E}\left[||\nabla_\psi\widehat{\mathcal{G}}_\mathcal{B}(\hat\eta^k, \psi^k, \theta^k) - \nabla_\psi\mathcal{G}_\mathcal{B}(\eta^{*,k}, \psi^k, \theta^k)||^2\right]$$

$$\leq 2\mathbb{E}\left[||\nabla_\psi\widehat{\mathcal{G}}_\mathcal{B}(\hat\eta^k, \psi^k, \theta^k) - \nabla_\psi\mathcal{G}_\mathcal{B}(\hat\eta^k, \psi^k, \theta^k)||^2\right] + 2\mathbb{E}\left[||\nabla_\psi\mathcal{G}_\mathcal{B}(\hat\eta^k, \psi^k, \theta^k) - \nabla_\psi\mathcal{G}_\mathcal{B}(\eta^{*,k}, \psi^k, \theta^k)||^2\right]$$

$$\leq 2H_\psi^2 C\mathcal{E}^k + 2\frac{L_{\psi\eta}^2 \cdot \zeta^k}{z^2}. \tag{B.57}$$

Finally, we can be obtained:

$$\mathbb{E}\left[\left\langle \nabla_\psi \mathcal{R}\left(\eta^{*,k}, \psi^k, \theta^k\right), \psi^{k+1} - \psi^k\right\rangle + \frac{L_\psi}{2}\left\|\psi^{k+1} - \psi^k\right\|^2\right]$$

$$\leq -\frac{\alpha_\psi}{2}\mathbb{E}\left[||\nabla_\psi \mathcal{R}\left(\eta^{*,k}, \psi^k, \theta^k\right)||^2\right] + \frac{\alpha_\psi}{2}(1 + L_\psi\alpha_\psi)(2H_\psi^2 V\mathcal{E}^k + 2\frac{L_{\psi\eta}^2 \cdot \zeta^k}{z^2}) + L_\psi\alpha_\psi^2\sigma_\psi^2. \tag{B.58}$$

For (b), similar to the proof for $\psi$ in **B.55**), for $\theta$, we can get:

$$\mathbb{E}\left[\left\langle \nabla_\theta \mathcal{R}\left(\eta^{*,k}, \psi^k, \theta^k\right), \theta^{k+1} - \theta^k\right\rangle + \frac{L_\psi}{2}\left\|\theta^{k+1} - \theta^k\right\|^2\right]$$

$$\leq -\frac{\alpha_\theta}{2}\mathbb{E}\left[||\nabla_\theta \mathcal{R}\left(\eta^{*,k}, \psi^k, \theta^k\right)||^2\right] + \frac{\alpha_\theta}{2}(1 + L_\theta\alpha_\theta)\mathbb{E}\left[||\hat\nabla_\theta\hat{\mathcal{G}}_\mathcal{B}(\hat\eta^k, \psi^k, \theta^k) - \nabla_\theta\mathcal{G}_\mathcal{B}(\eta^{*,k}, \psi^k, \theta^k)||^2\right]$$

$$+ L_\theta\alpha_\theta^2\mathbb{E}\left[||\nabla_\theta\mathcal{G}_\mathcal{B}(\eta^{*,k}, \psi^k, \theta^k) - \nabla_\theta\mathcal{R}\left(\eta^{*,k}, \psi^k, \theta^k\right)||^2\right]$$

$$+ \alpha_\theta(1 - L_\theta\alpha_\theta)\mathbb{E}\left[\nabla_\theta\mathcal{R}\left(\eta^{*,k}, \psi^k, \theta^k\right), \nabla_\theta\mathcal{R}\left(\eta^{*,k}, \psi^k, \theta^k\right) - \nabla_\theta\mathcal{G}_\mathcal{B}(\eta^{*,k}, \psi^k, \theta^k)\right]. \tag{B.59}$$

Under **Assumption 2**, we can get:

$$\mathbb{E}\left[nabla_\theta\mathcal{R}\left(\eta^{*,k}, \psi^k, \theta^k\right), \nabla_\theta\mathcal{R}\left(\eta^{*,k}, \psi^k, \theta^k\right) - \nabla_\theta\mathcal{G}_\mathcal{B}(\eta^{*,k}, \psi^k, \theta^k)\right] = 0 \tag{B.60}$$

$$\mathbb{E}\left[||\nabla_\theta\mathcal{G}_\mathcal{B}(\eta^{*,k}, \psi^k, \theta^k) - \nabla_\theta\mathcal{R}\left(\eta^{*,k}, \psi^k, \theta^k\right)||^2\right] \leq \sigma_\theta^2. \tag{B.61}$$

Furthermore, under **Lemma 3**, **Lemma 5** and **Lemma 7**, we can get:

$$\mathbb{E}\left[||\hat\nabla_\theta\hat{\mathcal{G}}_\mathcal{B}(\hat\eta^k, \psi^k, \theta^k) - \nabla_\theta\mathcal{G}_\mathcal{B}(\eta^{*,k}, \psi^k, \theta^k)||^2\right]$$

$$\leq 3\mathbb{E}\left[||\hat\nabla_\theta\widehat{\mathcal{G}}_\mathcal{B}(\hat\eta^k, \psi^k, \theta^k) - \nabla_\theta\widehat{\mathcal{G}}_\mathcal{B}(\hat\eta^k, \psi^k, \theta^k)||^2\right] + 3\mathbb{E}\left[||\nabla_\theta\widehat{\mathcal{G}}_\mathcal{B}(\hat\eta^k, \psi^k, \theta^k) - \nabla_\theta\mathcal{G}_\mathcal{B}(\hat\eta^k, \psi^k, \theta^k)||^2\right]$$

$$+ 3\mathbb{E}\left[||\nabla_\theta\mathcal{G}_\mathcal{B}(\hat\eta^k, \psi^k, \theta^k) - \nabla_\theta\mathcal{G}_\mathcal{B}(\eta^{*,k}, \psi^k, \theta^k)||^2\right]$$

$$\leq 3\frac{L^2\mu^2 d^2}{4} + 3Q_\theta^2 H_\theta^2 C\mathcal{E}^k + 3\frac{L_{\theta\eta}^2 \cdot \zeta^k}{z^2}. \tag{B.62}$$

Finally, we can be obtained:

$$\mathbb{E}\left[\left\langle \nabla_\psi \mathcal{R}\left(\eta^{*,k}, \psi^k, \theta^k\right), \psi^{k+1} - \psi^k\right\rangle + \frac{L_\psi}{2}\left\|\psi^{k+1} - \psi^k\right\|^2\right]$$

$$\leq -\frac{\alpha_\theta}{2}\mathbb{E}\left[||\nabla_\theta \mathcal{R}\left(\eta^{*,k}, \psi^k, \theta^k\right)||^2\right] + \frac{\alpha_\theta}{2}(1 + L_\theta\alpha_\theta)(3\frac{L^2\mu^2 d^2}{4} + 3Q_\theta^2 H_\theta^2 C\mathcal{E}^k + 3\frac{L_{\theta\eta}^2 \cdot \zeta^k}{z^2}) + L_\theta\alpha_\theta^2\sigma_\theta^2. \tag{B.63}$$

Substituting a) and b), we can get:

$$
\mathbb{E}\left[\mathcal{R}\left(\eta^{*,k+1},\psi^{k+1},\theta^{k+1}\right)-\mathcal{R}\left(\eta^{*,k},\psi^{k},\theta^{k}\right)\right]
$$
$$
\leq \mathbb{E}\left[\left\langle\nabla_{\psi}\mathcal{R}\left(\eta^{*,k},\psi^{k},\theta^{k}\right),\psi^{k+1}-\psi^{k}\right\rangle+\frac{L_{\psi}}{2}\left\|\psi^{k+1}-\psi^{k}\right\|^{2}\right]
$$
$$
+\mathbb{E}\left[\left\langle\nabla_{\theta}\mathcal{R}\left(\eta^{*,k},\psi^{k},\theta^{k}\right),\theta^{k+1}-\theta^{k}\right\rangle+\frac{L_{\theta}}{2}\left\|\theta^{k+1}-\theta^{k}\right\|^{2}\right]
$$
$$
\leq -\frac{\alpha_{\psi}}{2}\mathbb{E}\left[||\nabla_{\psi}\mathcal{R}\left(\eta^{*,k},\psi^{k},\theta^{k}\right)||^{2}\right]+\frac{\alpha_{\psi}}{2}(1+L_{\psi}\alpha_{\psi})(2H_{\psi}^{2}C\mathcal{E}^{k}+2\frac{L_{\psi\eta}^{2}\cdot\zeta^{k}}{z^{2}})+L_{\psi}\alpha_{\psi}^{2}\sigma_{\psi}^{2}
$$
$$
-\frac{\alpha_{\theta}}{2}\mathbb{E}\left[||\nabla_{\theta}\mathcal{R}\left(\eta^{*,k},\psi^{k},\theta^{k}\right)||^{2}\right]+\frac{\alpha_{\theta}}{2}(1+L_{\theta}\alpha_{\theta})(3\frac{L^{2}\mu^{2}d^{2}}{4}+3Q_{\theta}^{2}H_{\theta}^{2}C\mathcal{E}^{k}+3\frac{L_{\theta\eta}^{2}\cdot\zeta^{k}}{z^{2}})+L_{\theta}\alpha_{\theta}^{2}\sigma_{\theta}^{2}.
$$
$$(B.64)$$

Since $\psi$ and $\theta$ are updated synchronously in the outer loop, we take $\alpha_{m}=\min\{\alpha_{\psi},\alpha_{\theta}\}$, and combine the gradient:

$$
\mathbb{E}\left[\mathcal{R}\left(\eta^{*,k+1},\psi^{k+1},\theta^{k+1}\right)-\mathcal{R}\left(\eta^{*,k},\psi^{k},\theta^{k}\right)\right]
$$
$$
\leq -\frac{\alpha_{m}}{2}\mathbb{E}\left[||\nabla\mathcal{R}\left(\eta^{*,k},\psi^{k},\theta^{k}\right)||^{2}\right]+\frac{\alpha_{\psi}}{2}(1+L_{\psi}\alpha_{\psi})(2H_{\psi}^{2}C\mathcal{E}^{k}+2\frac{L_{\psi\eta}^{2}\cdot\zeta^{k}}{z^{2}})+L_{\psi}\alpha_{\psi}^{2}\sigma_{\psi}^{2}
$$
$$
+\frac{\alpha_{\theta}}{2}(1+L_{\theta}\alpha_{\theta})(3\frac{L^{2}\mu^{2}d^{2}}{4}+3Q_{\theta}^{2}H_{\theta}^{2}C\mathcal{E}^{k}+3\frac{L_{\theta\eta}^{2}\cdot\zeta^{k}}{z^{2}})+L_{\theta}\alpha_{\theta}^{2}\sigma_{\theta}^{2}.
$$
$$(B.65)$$

Summing these inequalities from $k=0$ to $\mathcal{K}-1$, take $\mathcal{E}=\underset{k=0,\ldots,K-1}{max}(\mathcal{E}^{k})$, and then $\zeta=\underset{k=0,\ldots,K-1}{max}(\zeta^{k})$:

$$
\frac{1}{\mathcal{K}}\sum_{k=0}^{\mathcal{K}-1}\frac{\alpha_{m}}{2}\mathbb{E}\left[||\nabla\mathcal{R}\left(\eta^{*,k},\psi^{k},\theta^{k}\right)||^{2}\right]
$$
$$
\leq \frac{1}{\mathcal{K}}\sum_{k=0}^{\mathcal{K}-1}\mathbb{E}\left[\mathcal{R}\left(\eta^{*,k},\psi^{k},\theta^{k}\right)-\mathcal{R}\left(\eta^{*,k+1},\psi^{k+1},\theta^{k+1}\right)\right]+\frac{\alpha_{\psi}}{2}(1+L_{\psi}\alpha_{\psi})(2H_{\psi}^{2}C\mathcal{E}+2\frac{L_{\psi\eta}^{2}\cdot\zeta}{z^{2}})+L_{\psi}\alpha_{\psi}^{2}\sigma_{\psi}^{2}
$$
$$
+\frac{\alpha_{\theta}}{2}(1+L_{\theta}\alpha_{\theta})(3\frac{L^{2}\mu^{2}d^{2}}{4}+3Q_{\theta}^{2}H_{\theta}^{2}C\mathcal{E}+3\frac{L_{\theta\eta}^{2}\cdot\zeta}{z^{2}})+L_{\theta}\alpha_{\theta}^{2}\sigma_{\theta}^{2}
$$
$$
=\mathbb{E}\left[\mathcal{R}\left(\eta^{*,0},\psi^{0},\theta^{0}\right)-\mathcal{R}\left(\eta^{*,\mathcal{K}},\psi^{\mathcal{K}},\theta^{\mathcal{K}}\right)\right]+\frac{\alpha_{\psi}}{2}(1+L_{\psi}\alpha_{\psi})(2H_{\psi}^{2}C\mathcal{E}+2\frac{L_{\psi\eta}^{2}\cdot\zeta}{z^{2}})+L_{\psi}\alpha_{\psi}^{2}\sigma_{\psi}^{2}
$$
$$
+\frac{\alpha_{\theta}}{2}(1+L_{\theta}\alpha_{\theta})(3\frac{L^{2}\mu^{2}d^{2}}{4}+3Q_{\theta}^{2}H_{\theta}^{2}C\mathcal{E}+3\frac{L_{\theta\eta}^{2}\cdot\zeta}{z^{2}})+L_{\theta}\alpha_{\theta}^{2}\sigma_{\theta}^{2}.
$$
$$(B.66)$$

Then, we define $\Lambda=\mathcal{R}\left(\eta^{*,0},\psi^{0},\theta^{0}\right)-\inf_{k}(\mathcal{R}\left(\eta^{*,k},\psi^{k},\theta^{k}\right))$ and $\alpha_{M}=\max\{\alpha_{\psi},\alpha_{\theta}\}$:

$$\frac{1}{\mathcal{K}} \sum_{k=0}^{\mathcal{K}-1} \mathbb{E}\left[||\nabla \mathcal{R}\left(\eta^{*,k}, \psi^k, \theta^k\right)||^2\right]$$

$$\leq \frac{2\Lambda}{\alpha_m \mathcal{K}} + \frac{\alpha_\psi(1 + L_\psi \alpha_\psi)}{\alpha_m}(2H_\psi^2 C\mathcal{E} + 2\frac{L_{\psi\eta}^2 \cdot \zeta}{z^2}) + \frac{2L_\psi \alpha_\psi^2 \sigma_\psi^2}{\alpha_m}$$

$$+ \frac{\alpha_\theta(1 + L_\theta \alpha_\theta)}{\alpha_m}(3\frac{L^2\mu^2 d^2}{4} + 3Q_\theta^2 H_\theta^2 C\mathcal{E} + 3\frac{L_{\theta\eta}^2 \cdot \zeta}{z^2}) + \frac{2L_\theta \alpha_\theta^2 \sigma_\theta^2}{\alpha_m}$$

$$\leq \frac{2\Lambda}{\alpha_m \mathcal{K}} + \frac{\alpha_M(1 + L_\psi \alpha_M)}{\alpha_m}(2H_\psi^2 C\mathcal{E} + 2\frac{L_{\psi\eta}^2 \cdot \zeta}{z^2}) + \frac{2L_\psi \alpha_M^2 \sigma_\psi^2}{\alpha_m}$$

$$+ \frac{\alpha_M(1 + L_\theta \alpha_M)}{\alpha_m}(3\frac{L^2\mu^2 d^2}{4} + 3Q_\theta^2 H_\theta^2 C\mathcal{E} + 3\frac{L_{\theta\eta}^2 \cdot \zeta}{z^2}) + \frac{2L_\theta \alpha_M^2 \sigma_\theta^2}{\alpha_m}$$

$$= (\frac{2\Lambda}{\alpha_m \mathcal{K}} + \frac{2L_\psi \alpha_M^2 \sigma_\psi^2}{\alpha_m} + \frac{2L_\theta \alpha_M^2 \sigma_\theta^2}{\alpha_m}) + \frac{\alpha_M}{\alpha_m}\left[2(1 + L_\psi \alpha_M)H_\psi^2 C\mathcal{E} + 3(1 + L_\theta \alpha_M)Q_\theta^2 H_\theta^2 C\mathcal{E}\right]$$

$$+ \frac{3\alpha_M(1 + L_\theta \alpha_M)L^2\mu^2 d^2}{4\alpha_m} + \frac{\alpha_M \tau}{\alpha_m z^2}\left[2(1 + L_\psi \alpha_M)L_{\psi\eta}^2 + 3(1 + L_\theta \alpha_M)L_{\theta\eta}^2\right]$$

$$= (\frac{2\Lambda}{\alpha_m \mathcal{K}} + \frac{2L_\psi \alpha_M^2 \sigma_\psi^2}{\alpha_m} + \frac{2L_\theta \alpha_M^2 \sigma_\theta^2}{\alpha_m}) + \mathcal{E}\frac{\alpha_M}{\alpha_m}\left[2(1 + L_\psi \alpha_M)H_\psi^2 C + 3(1 + L_\theta \alpha_M)Q_\theta^2 H_\theta^2 C\right]$$

$$+ \mu^2 \frac{3\alpha_M(1 + L_\theta \alpha_M)L^2 d^2}{4\alpha_m} + \frac{\alpha_M \tau}{\alpha_m z^2}\left[2(1 + L_\psi \alpha_M)L_{\psi\eta}^2 + 3(1 + L_\theta \alpha_M)L_{\theta\eta}^2\right]$$

$$= (\frac{2\Lambda}{\alpha_m \mathcal{K}}) + (\frac{2L_\psi \alpha_M^2 \sigma_\psi^2}{\alpha_m} + \frac{2L_\theta \alpha_M^2 \sigma_\theta^2}{\alpha_m})$$

$$+ \mathcal{E}\frac{\alpha_M}{\alpha_m}\left[2(1 + L_\psi \alpha_M)H_\psi^2 C + 3(1 + L_\theta \alpha_M)Q_\theta^2 H_\theta^2 C + \frac{3H_\theta^2 C\pi(M,N)}{z^2}(2(1 + L_\psi \alpha_M)L_{\psi\eta}^2 + 3(1 + L_\theta \alpha_M)L_{\theta\eta}^2)\right]$$

$$+ \mu^2(\frac{3\alpha_M(1 + L_\theta \alpha_M)L^2 d^2}{4\alpha_m} + \frac{3\pi(M,N)L^2 d^2 a_M(1 + L_\psi \alpha_M)L_{\psi\eta}^2}{2a_m z^2} + \frac{9\pi(M,N)L^2 d^2 a_M(1 + L_\theta \alpha_M)L_{\theta\eta}^2}{4a_m z^2})$$

$$+ \frac{3\alpha_M K^2 \pi(M,N)}{\alpha_m z^2}\left[2(1 + L_\psi \alpha_M)L_{\psi\eta}^2 + 3(1 + L_\theta \alpha_M)L_{\theta\eta}^2\right]. \tag{B.67}$$

**Corollary 1** According to **Theorem 1**:If we choose $\alpha_\theta$ and $\alpha_\psi$ as $\mathcal{O}(\frac{1}{\sqrt{\mathcal{K}}})$, $\mu = \mathcal{O}(\frac{1}{\mathcal{K}^{\frac{1}{4}}})$, $\mathcal{E} = \mathcal{O}(\frac{1}{\sqrt{\mathcal{K}}})$, $\Gamma = \mathcal{O}(\frac{1}{\sqrt{\mathcal{K}}})$, we can derive the sublinear convergence rate:

$$\frac{1}{\mathcal{K}} \sum_{k=0}^{\mathcal{K}-1} \mathbb{E}\left[||\nabla \mathcal{R}\left(\eta^{*,k}, \psi^k, \theta^k\right)||^2\right] \leq \mathcal{O}(\frac{1}{\sqrt{\mathcal{K}}}) + \mathcal{O}(\frac{N\mathcal{M}_K}{M}). \tag{B.68}$$

**Proof:**

$$\frac{1}{\mathcal{K}} \sum_{k=0}^{\mathcal{K}-1} \mathbb{E}\left[||\nabla \mathcal{R}\left(\eta^{*,k}, \psi^k, \theta^k\right)||^2\right]$$

$$\leq \mathcal{O}(\frac{1}{\sqrt{\mathcal{K}}})[(2\Lambda) + (2L_\psi \sigma_\psi^2 + 2L_\theta \sigma_\theta^2)$$

$$+ 2(1 + L_\psi \alpha_M)H_\psi^2 C + 3(1 + L_\theta \alpha_M)Q_\theta^2 H_\theta^2 C + \frac{3H_\theta^2 C\pi(M,N)}{z^2}(2(1 + L_\psi \alpha_M)L_{\psi\eta}^2 + 3(1 + L_\theta \alpha_M)L_{\theta\eta}^2)$$

$$+ (\frac{3\alpha_M(1 + L_\theta \alpha_M)L^2 d^2}{4\alpha_m} + \frac{3\pi(M,N)L^2 d^2 a_M(1 + L_\psi \alpha_M)L_{\psi\eta}^2}{2a_m z^2} + \frac{9\pi(M,N)L^2 d^2 a_M(1 + L_\theta \alpha_M)L_{\theta\eta}^2}{4a_m z^2})]$$

$$+ \frac{3K^2 \pi(M,N)}{z^2}\left[2(1 + L_\psi \alpha_M)L_{\psi\eta}^2 + 3(1 + L_\theta \alpha_M)L_{\theta\eta}^2\right]$$

$$= \mathcal{O}(\frac{1}{\sqrt{\mathcal{K}}}) + \mathcal{O}(\frac{N\mathcal{M}_K}{M}). \tag{B.69}$$

## C  Experiment Details

### C.1  Dataset details

Our experiments were constricted on public datasets MNIST and CIFAR10:

- **MNIST** [34]: A benchmark dataset for image classification, comprising 60,000 examples for training and 10,000 examples for testing.
- **CIFAR10** [31]: Another public dataset for image classification that consists of 60,000 images categorized into 10 classes.
- **CIFAR100** [31]: Another public dataset for image classification that consists of 60,000 images categorized into 100 classes.
- **Tiny-ImageNet** [33]: A scaled-down version of ImageNet that contains 200 classes with 500 training images, 50 validation images, and 50 test images per class. Each image has a resolution of 64×64 pixels, providing a more challenging benchmark than MNIST and CIFAR datasets.

To simulate the VFL scenario, we allocated distinct features to each party based on the described methodology in prior works [48, 53, 17]. We partition the last dimension of the features according to the feature proportion of each client. We use masking to ensure that each client receives distinct features.

### C.2  Adversarial attack

To validate robustness, we employed a suite of adversarial attack methods. FGSM is a fast, non-iterative attack [32]; PGD-$r$ iteratively perturbs input data using gradient information to maximize the model's loss [49]; and CW uses a custom loss function to ensure minimal perturbations while achieving misclassification [4]. CERTIFY (CER) generates adversarial perturbations with Gaussian noise [11]. For black-box attacks, we combined adversarial methods with zeroth-order optimization: FGSM (ZO-FGSM) and PGD (ZO-PGD) [8]. We also considered scenarios involving a third-party adversary, corrupting embeddings using different client selection strategies, including Thompson Sampling with Empirical Maximum Reward (E-TS) [16] and All Corruption Patterns (ALL).

### C.3  Hyperparameters

For the parameter updates of both the server and client models, we have adopted the Adam optimizer with a uniform learning rate of $\alpha_\psi = \alpha_\theta = 0.0001$.

Moreover, We follow the hyperparameters choices of [4, 12, 32, 55, 76, 82] for training.

Table 11: Hyperparameters for Adv. Training

| Dataset | Client Model | batch size | ZOO | | Compress | | Adv. | | DecVFAL | | PGD | FreeAT | FreeLB |
|---|---|---|---|---|---|---|---|---|---|---|---|---|---|
| | | | $q$ | $\mu$ | type | bit | $\epsilon$ | $\sigma$ | m | n | n | n | n |
| MNIST | MLP | 32 | 100 | 0.05 | scale | 2 | 0.02 | 0.002 | 5 | 10 | 40 | 8 | 40 |
| CIFAR10 | ResNet-18 | 80 | 200 | 0.5 | scale | 2 | 8/255 | 1/255 | 6 | 2 | 10 | 8 | 10 |
| MNIST | ResNet-18 | 32 | 100 | 0.05 | scale | 2 | 0.3 | 0.35 | 6 | 8 | 40 | 8 | 40 |
| CIFAR100 | ResNeXt-50 | 80 | 1000 | 0.5 | scale | 8 | 3/255 | 0.4/255 | 5 | 3 | 10 | 8 | 10 |

Table 12: Hyperparameters for Attack

| Dataset | Client Model | ZOO | | FGSM | PGD | | | CW | | | CER | ZO-FGSM | ZO-PGD | | | ALL & E-TS | | |
|---|---|---|---|---|---|---|---|---|---|---|---|---|---|---|---|---|---|---|
| | | $q$ | $\mu$ | $\epsilon$ | n | $\epsilon$ | $\sigma$ | n | $\sigma$ | c | $\epsilon$ | $\epsilon$ | n | $\epsilon$ | $\sigma$ | n | $\epsilon$ | $\sigma$ |
| MNIST | MLP | 100 | 0.05 | 16/255 | 40 | 24/255 | 4/255 | 100 | 0.32 | 0.5 | 128/255 | 64/255 | 40 | 96/255 | 12/255 | 10 | 96/255 | 12/255 |
| CIFAR10 | ResNet-18 | 200 | 0.05 | 0.01 | 10 | 8/255 | 1/255 | 100 | 128/255 | 0.8 | 64/255 | 32/255 | 40 | 32/255 | 2/255 | 10 | 32/255 | 64/255 |
| MNIST | ResNet-18 | 100 | 0.05 | 96/255 | 40 | 64/255 | 2/255 | 100 | 0.8 | 0.5 | 204/255 | 64/255 | 40 | 153/255 | 16/255 | 40 | 128/255 | 16/255 |
| CIFAR100 | ResNeXt-50 | 100 | 0.05 | 1/255 | 10 | 1/255 | 0.2/255 | 100 | 3/255 | 0.8 | 3/255 | 16/255 | 10 | 3/255 | 0.4/255 | 10 | 16/255 | 2/255 |

### C.4  Environment

In our experiments, we utilized the following software environment: PyTorch version 2.2.1, CUDA version 12.1, and Python version 3.11. The hardware specifications are detailed in Table 13.

Table 13: Hardware Specifications

| Experiment Description | CPU | GPU |
|---|---|---|
| MNIST Robust Training | AMD EPYC 7551P | A4000*1 |
| CIFAR-10 Robust Training | AMD EPYC 7452 | 4090*4 |
| CIFAR-100 Robust Training | AMD EPYC 7452 | 4090*4 |
| Performance across various NN architectures | Intel E5-2683 v4 | 4090*1 |
| Impact of split position | AMD EPYC 7J13 | 4090*4 |
| Impact of the number of modules | AMD EPYC 7J13 | 4090*4 |
| Impact of the number of the clients | Intel Platinum 8336C | 4090*8 |
| Limitation of the setting of M and N | Intel Fold 6430 | 4090*8 |

## C.5 Performance across various NN architectures.

We expanded our experiments by incorporating ResNet18 on the MNIST dataset, introducing a different architectural context for evaluating our framework. Similar as experiments in CIFAR-10, the entire model is partitioned into three modules: the first layers of the client models, the remaining layers of the client models, and the server's single-layer perceptron. As shown in Table 14, DecVFAL achieves the best robust performance while requiring only one-seventh of the training time per epoch for PGD adversarial training.

Table 14: Results of MNIST Robust Training with Resnet-18

| Training Methods | Clean Accuracy | White-Box Adv. Atk | | | Black-Box Adv. Atk | | | Third Adv. | | Train Time |
|---|---|---|---|---|---|---|---|---|---|---|
| | | FGSM | PGD | CW | CER | ZO-FGSM | ZO-PGD | ALL | E-TS | (s/epoch) |
| None | 98.66 | 56.56 | 12.26 | 20.99 | 68.37 | 48.67 | 71.43 | 47.96 | 75.71 | **90.86** |
| PGD | 98.23 | 84.73 | 74.74 | 21.16 | 83.98 | 38.21 | 83.69 | 44.56 | 72.49 | 1180.74 |
| FreeAT | 98.44 | 79.47 | 82.20 | **52.33** | 90.82 | **94.84** | 89.04 | 46.52 | 70.15 | 332.63 |
| FreeLB | 98.82 | 70.81 | 40.68 | 31.53 | 80.05 | 36.51 | 81.27 | 65.91 | 87.07 | 1419.10 |
| YOPO | 98.72 | 83.11 | 82.13 | 21.30 | 87.44 | 54.13 | 87.33 | **71.05** | **88.98** | 240.62 |
| DP | 98.63 | 80.20 | 66.63 | 29.80 | 80.63 | 42.68 | 81.55 | 44.77 | 70.31 | 1175.02 |
| MP | 98.12 | 81.38 | 74.47 | 36.31 | 86.42 | 53.64 | 84.84 | 50.38 | 74.10 | 1181.69 |
| Asy-PGD | 98.05 | 79.42 | 76.27 | 27.57 | 85.93 | 42.63 | 84.71 | 57.95 | 80.16 | 1167.09 |
| DecVFAL | **98.98** | **89.00** | **83.20** | 50.80 | **93.91** | 90.95 | **91.17** | 70.42 | 84.14 | **167.89** |

## C.6 Experiments on Tiny-ImageNet

To evaluate the scalability of DecVFAL to more complex datasets, we conducted experiments on Tiny-ImageNet using similar AT methods. Table 15 presents the robustness performance and training time across different perturbation budgets. DecVFAL-3-2 consistently outperforms FastAT and FreeAT, achieving robust accuracy comparable to or better than ATAS while requiring less training time. Notably, DecVFAL's performance is close to standard PGD-5 with approximately half the training time. These results demonstrate that DecVFAL maintains its computational advantages and robustness benefits when applied to more challenging datasets with higher resolution images and increased class complexity, further confirming its practicality for real-world deployments.

Table 15: Experimental results on Tiny-ImageNet

| Method | Time (h) | $\varepsilon = 1/255$ | | $\varepsilon = 2/255$ | | $\varepsilon = 4/255$ | |
|---|---|---|---|---|---|---|---|
| | | PGD-5 | AA | PGD-5 | AA | PGD-5 | AA |
| PGD-5 | 8.03 | **34.17** | **32.89** | **28.39** | **24.14** | 16.48 | **11.47** |
| FreeAT-4 | 7.53 | 29.45 | 27.78 | 21.67 | 16.66 | 9.27 | 4.97 |
| FastAT | 4.64 | 24.04 | 22.42 | 19.45 | 15.27 | 10.52 | 6.18 |
| ATAS | 5.22 | 32.83 | 31.57 | 27.17 | 22.24 | 16.25 | 10.25 |
| DecVFAL-3-2 | **4.30** | 33.41 | 31.85 | 27.95 | 23.11 | **16.74** | 10.93 |

