# OpenReview forum: "Accelerated Vertical Federated Adversarial Learning through Decoupling Layer-Wise Dependencies"
_NeurIPS.cc/2025/Conference — NeurIPS 2025 poster_

### Official Review · Reviewer_Ecn8 · 2025-07-01

**Clarity:** 3
**Significance:** 2
**Originality:** 3
**Rating:** 4
**Confidence:** 2

**Summary:**

This paper proposes DecVFAL framework to accelerate vertical federated adversarial training. DecVFAL decouples the bottom modules of clients and backpropagation to eliminate idle waiting times. Experiments on four public benchmarks demonstrate that DecVFAL effectively balances computational efficiency with model robustness.

**Questions:**

1. Will deploying more complex models on the server side, such as multi-layer perceptron affect the results?
2. Typo: "single-layer perception" in line 859

**Ethical Concerns:**

["NO or VERY MINOR ethics concerns only"]

**Final Justification:**

The rebuttal effectively addresses the main concerns, particularly on client scaling and server model analysis. Therefore, I maintain my score of 4.

**Limitations:**

yes

**Quality:**

3

**Strengths And Weaknesses:**

- Strengths
1. The paper is well-written and relatively easy to follow.
2. There are extensive theoretical analysis about the convergence analysis.
3. Well-designed ablation studies of the impacts of components and hyperparameters are conducted.

- Weaknesses
1. The problem setting seems niche with only 2 clients in main experiments. Even in ablation studies, the max number of clients is only 7.
2. The model deployed in server side is kept as a single-layer perceptron. Analysis or justification for such choice could be helpful.
3. The speedup is dependent on the split position. Training time increases with more layers included in bottom layers, which restricts the architecture.

---

> ### Author Rebuttal · Authors · 2025-07-31
>
> Thank you for your insightful comments and for recognizing the contribution of our work. Your feedback has been invaluable in enhancing the quality and clarity of our manuscript. We reply to the weaknesses and questions.
>
> **W1-Client Scale:**
>
> **Experiments on More Clients.** Thank you for this valuable feedback. We have conducted additional experiments on MNIST with **14 and 28 clients** as shown in Table 1. These results demonstrate that DecVFAL maintains strong robustness and computational efficiency across different client configurations.
>
> **Table 1: Results with Varying Client Numbers (MNIST)**
>
> | No. Clients | Method | Clean | Robust  |Accuracy|| Time (h) &darr;| Speedup |
> |-------------|--------|---------------|-------------------|------------|---------|-|-|
> | | | | FGSM | PGD | AA |  | vs PGD |
> | 14 | PGD  | 96.79 | 86.33 | 85.16 | 78.31 | 3.09 | 1× |
> | | FreeAT  | **97.72** | 65.10 | 67.50 | 47.48 | 2.47 | 1.25× |
> | | YOPO  | 96.74 | 83.64 | 83.19 | 74.50 | 1.77 | 1.74× |
> | | DecVFAL  | 97.11 | **89.63** | **86.40** | **79.44** | **0.71** | **4.35×** |
> | 28 | PGD | 95.95 | 88.53 | 87.16 | 80.27 | 3.59 | 1× |
> | | FreeAT | **96.83** | 82.39 | 83.00 | 65.92 | 0.87 | 4.12× |
> | | YOPO | 95.93 | 89.09 | **88.86** | **83.75** | 0.83 | 4.32× |
> | | DecVFAL | 96.00 | **89.99** | 88.04 | 82.63 | **0.73** | **4.92×** |
>
> **W2&Q1-Server model:**
>
> 1. It is worth that server and client represent functional roles rather than physical deployment constraints.
> The single-layer perceptron for the server model is a **standard configuration in VFL research**, following established VFL frameworks where the server typically handles final classification while clients process feature-specific computations [1-3].
>
>
> 2. Thank you for this insightful question. We tested DecVFAL (M=6, N=8)  on the MNIST dataset by deploying various server-side models: 1-layer perceptrons, 4-layer perceptrons, and 16-layer perceptrons. The experimental results reveal that as the server model becomes more complex (with more layers), it tends to slow down the entire framework's parallel efficiency. This happens because the server acts as a single module, and when it takes longer to process, **all other modules must wait for its output before proceeding**.
>
> **Table: Results with Different Server Model Complexities**
>
> | Server Model | Method | Clean | Robust | Accuracy|| Time &darr; |
> |--------------|--------|---------------|-------------------|------------|-|---------|
> | | | | FGSM | PGD | AA | (s/epoch) |
> | 1-Layer | DecVFAL | 98.10 | 90.40 | **91.93** | **89.58** | **9.866** |
> | 4-Layer  | DecVFAL | **98.35** | **90.57** | 91.70 | 88.74 | 9.986 |
> | 16-Layer | DecVFAL | 97.75 | 89.78 | 89.60 | 83.97 | 13.218 |
>
>
>
> **W3-Split Position:**
>
> We acknowledge this important practical consideration. When one module contains significantly more layers, other modules must idle waiting for its output, reducing overall efficiency as demonstrated in Table 6. However, our framework provides flexibility to optimize split positions based on computational balance. We recommend treating server-client communication as a module boundary and ensuring each module's computation time is **balanced and slightly greater than communication delay**. This design enables modules to simultaneously send embeddings while receiving gradients, effectively **eliminating communication-induced idle time and achieving maximum acceleration benefits**.
>
>
> **Q2-Typo:**
>
> Thank you for catching this typo. We will correct "single-layer perception" to "single-layer perceptron" in line 859.
>
>
> [1] LIU, Yang, et al. Vertical federated learning: Concepts, advances, and challenges. IEEE transactions on knowledge and data engineering, 2024, 36.7: 3615-3634.
>
> [2] WEI, Kang, et al. Vertical federated learning: Challenges, methodologies and experiments. arXiv preprint arXiv:2202.04309, 2022.
>
> [3] WANG, Ganyu, et al. A unified solution for privacy and communication efficiency in vertical federated learning. Advances in Neural Information Processing Systems, 2023, 36: 13480-13491.

---

> > ### Comment · Reviewer_Ecn8 · 2025-08-06
> >
> > Thanks for the author's clarification. My concerns have been addressed.

---

### Official Review · Reviewer_2ipB · 2025-07-02

**Clarity:** 3
**Significance:** 2
**Originality:** 1
**Rating:** 4
**Confidence:** 4

**Summary:**

The paper proposes DecVFAL, a fast adversarial learning technique for vertical federated learning setting. To accelerate training:

1.	Client-side performs lazy refinement steps on the small bottom module ‘N’ times while keeping the upper-layer gradients frozen.
2.	Use delayed gradients concurrently to update the rest of the modules.

Authors provide theoretical guarantees for convergence and demonstrate 3-10x speed-up on various datasets.

**Questions:**

1.	Can you briefly instate the significance of AL in VFL and why does it need acceleration in practical scenarios? Considering most of the applications (e.g., in hospitals) involves collaboration of a few parties and would have enough resources.

2.	In this VFL scenario, a separate server is utilized between parties and assumed to hold the labels. Is this a realistic in VFL? What other threats is this setup exposed to? How would the method adapt when parties have non overlapping user sets and rely on secure entity matching?

**Ethical Concerns:**

["NO or VERY MINOR ethics concerns only"]

**Final Justification:**

My doubts about the experimental settings are clear now and the additional results and discussion on my comments seems satisfactory. However, considering most of the gains are marginal for bigger datasets (almost 2-3x), I am increasing my score by 1 point.

**Limitations:**

Yes some of them.

**Paper Formatting Concerns:**

Figure 5 is in fact a table and also wrongly referred in the Section 6.5.

DecVFAL-3-3 and DecVFAL-6-2 used without proper explanation.

**Quality:**

2

**Strengths And Weaknesses:**

**Strengths**:
Although the technique comprises well-known tricks (gradient reuse, async updates), it shows good practical speed-up for training. Provides extensive comparisons with baselines with ablations on number of clients, modules, and other decoupling hyperparameters.

**Weaknesses**:

1.	Novelty is marginal as technique largely reuses know ideas in this context.

2.	Most of the compared baselines are outdated (mostly from 2019) and some recent ones (e.g., FastAT, ATAS) are only lightly covered.

3.	Networks that are used are small (e.g., 84 % clean accuracy on CIFAR 10). It is unclear if results hold for deeper models.

4.	DecVFAL sometimes yields higher robust accuracy than full PGD despite using weaker inner maximisers with small module in the client-side. It indeed entails strong clarification and justifications. Even the theoretical analysis suggest degradation in performance due to the added bias.

5.	The results are inconsistent. For instance, why does PGD and FreeLB accuracy trend differs strongly in Figure 3 and Figure 4 for different datasets?

6.	The results for impact of split position in Table 6 seems counter intuitive. Large bottom modules providing better gradient suggests increase in performance.

7.	Experiments max out at seven clients and assume perfectly aligned sample IDs. Larger client size or heterogeneous environments are untested.

8.	The sensitivity of the parameters M and N should be carefully studied and discussed.

9.	The analysis of the communication cost is missing. Using decoupled modules suggest substantial increase in communication of smashed layer’s embeddings and should be quantified.

---

> ### Author Rebuttal · Authors · 2025-07-31
>
> We sincerely thank you for your insightful comments and constructive criticisms. Your feedback has been invaluable in improving the quality and clarity of our manuscript. Below, we address the weaknesses and respond to your questions.
>
> **W1-Novelty:**
>
> Due to space limitations, please refer to W3 of Reviewer ikWZ.
>
> **W2-Baselines:**
>
> 1.  We selected sota AT acceleration methods (PGD, FreeAT, YOPO). They specifically target iteration complexity reduction, **directly addressing the primary bottleneck in VFL scenarios: communication overhead**. These baselines provide the most relevant comparison for communication-constrained environments like VFAL.
>
> 2. While we recognize recent AT developments, **most focus on improving robustness** through sample optimization (DOM [1]) or loss function enhancements (MART [2]) rather than addressing the computational bottleneck that VFAL faces. These approaches tackle complementary but different challenges.
>
> 3. Following your valuable suggestion, we enhanced our CIFAR-10 experiments to include recent methods like DOM [1] and MART [2] in Table 1. **Importantly, our decoupling framework is compatible with these optimization strategies**, allowing for combined benefits as shown in our "DecVFAL-6-2+DOM" results.
>
> **Table 1: Comparison with Recent AT Methods on CIFAR-10**
>
> |Method|Clean|4/255||8/255||12/255||Time&darr;|Speedup|
> |-|-|-|-|-|-|-|-|-|-|
> |||PGD|AA|PGD|AA|PGD|AA|(h)|(vsPGD)|
> |PGD|78.00|68.47|63.08|44.61|34.55|34.59|16.29|8.23|1.00×|
> |DOM[1]|75.06|63.74|62.26|44.26|36.59|36.87|21.29|13.26|0.62×|
> |MART[2]|80.98|64.91|**63.76**|43.22|32.46|31.75|14.28|7.85|1.05×|
> |DecVFAL-6-2|**81.83**|**68.59**|62.34|**46.27**|36.48|36.14|18.72|2.75|3.00×|
> |DecVFAL-6-2+DOM|74.31|66.22|61.44|46.24|**39.16**|**39.17**|**21.97**|3.47|2.37×|
> |DecVFAL-3-3|81.24|66.19|60.58|42.96|31.76|31.98|15.11|**1.45**|**5.68×**|
>
>
> **W3-Deeper model:**
>
> 1. **Standard Benchmarks for Comparison.** Our experimental settings follow well-established protocols from VFL research [6-7] and adversarial training studies (e.g., RobustBench [4]). This ensures fair comparison with existing methods and enables reproducible results.
>
> 2. **Deeper Architectures Included.** We have actually evaluated DecVFAL on several deeper models, including **ResNeXt-50 on CIFAR-100 and Tiny-ImageNet** (detailed in Section 6.1 and Appendix C.6). Since DecVFAL is designed to be architecture-agnostic, it naturally scales to modern deep networks without structural modifications.
>
> 3. **Additional Experiments.** We are currently conducting experiments with deeper networks on CIFAR-10 to further validate our approach. These results will be available within one week.
>
> **W4-PGD Performance:**
>
> The key insight is that DecVFAL performs significantly **more adversarial updates ($M\times N$) compared to standard PGD ($r$ iterations)**, where typically $M\times N > r$ in our experiments (e.g., 5$\times$10=50 vs. 40 updates for MNIST, 6$\times$2=12 vs. 10 updates for CIFAR10). While individual updates use weaker gradients, the substantially **higher update frequency provides more comprehensive adversarial space exploration**, compensating for per-update approximation errors.
>
>
> **W5-Performance differences:**
>
> The primary reason for the accuracy differences is the **different adversarial sample generation settings**: **40 iterations** on MNIST versus **10 iterations** on CIFAR-10 to generate adversarial samples.
> 1. **Slower Convergence for PGD.** The high iteration count combined with MNIST's **2-layer perceptron model** results in **slower convergence** compared to others.
> 2. **Overfitting for FreeLB**:  FreeLB's **gradient accumulation mechanism** combined with 40 iterations causes **overfitting issues**, leading to poor generalization and **accuracy degradation**.
>
>
> **W6-Number of Layers in Bottom Module:**
>
> This counter-intuitive result in Table 6 is caused by **error accumulation during backpropagation**. While larger bottom modules do provide better gradient approximations and reduce the initial delayed gradient error, **this error still persists** and becomes progressively **larger** as it propagates backward through the network layers. This **accumulation effect** ultimately leads to the unstable performance we observed.
>
> **W7-Client Size and Heterogeneous Environments:**
>
> 1. **Experiments on More Clients.** We have conducted additional experiments on MNIST with **14 and 28 clients** as shown in Table 1 in W1 of Reviewer Ecn8.
> 2. **About Heterogeneous Settings.** **Sample ID alignment is indeed a separate challenge in VFL** that involves entity resolution and record linkage [5]. Our work focuses on the complementary problem of speeding up adversarial training when participants already have aligned data, following standard VFL assumptions.
>
>
> **W8-The selection of M and N:**
>
> Due to space limitations, please refer to W4 of Reviewer yT3W.
>
> **W9-Communication:**
>
> 1. Decoupling does **not introduce additional communication overhead**, as only the server module and the top-layer client modules require true server-client communication, while other inter-module communications are purely local computations that barely affect efficiency.
> 2. **To fully address your concern, we have included detailed communication cost statistics below** to quantify the actual communication requirements and demonstrate the efficiency of our approach.
>
> **Table 2: Communication Cost**
>
> |Methods|Comm.Cost|(MB)&darr;||Speedup (vs PGD)|
> |-|-|-|-|-|
> ||MNIST|CIFAR10|CIFAR100||
> |PGD|294.48|122.92|261.86|1.00×|
> |FreeLB|294.48|122.92|261.86|0.98×|
> |FreeAT|117.79|98.34|209.49|1.18×|
> |YOPO|147.24|**61.46**|**130.93**|0.99×|
> |DecVFAL|**73.62**|73.75|**130.93**|**3.00×**|
>
> **Q1-Why AT and Acceleration are needed:**
>
> 1. **Why AT is essential in VFL.** VFL faces unique security challenges as detailed in Appendix A.3. These include **third-party attackers modifying embeddings** during communication and **malicious participants** corrupting local features. While we could use different defenses for each threat, **AT offers a unified solution** that builds inherent robustness against multiple attack types simultaneously.
>
> 2. **The resource utilization challenge.** You're absolutely right that institutions like hospitals have substantial computational resources. However, this actually shifts the bottleneck to **how efficiently we use these resources**. In VFAL, generating each adversarial sample requires multiple rounds of **client-server communication for gradient updates**. This creates **significant idle time** where powerful computing resources sit unused.
>
> 3. Our approach reduces communication rounds from r to M, allowing modules to **process computations in parallel during communication delays**. This means institutions can fully utilize their existing computational infrastructure rather than leaving it idle. Our experiments show **consistent 3-10× speedup**.
>
> **Q2-VFL setup:**
>
> 1. The server-holding-labels setup is **a standard and realistic assumption** in VFL literature, as evidenced by numerous works [3,6,7], where one party naturally possesses the target labels. An example is fraud detection where a bank holds transaction labels (whether fraud) and transaction history feature while partnering with e-commerce platforms holding user shopping features.
> 2. It is worth that 'server' and 'client' are **roles within the VFL framework** rather than fixed physical entities—in practical deployments, these roles can be **distributed across different participating organizations** based on their data ownership and computational capabilities, with the label-holding participant typically assuming the server role for coordination purposes.
> 3. Our threat model considers multiple adversarial attack vectors including **third-party adversaries** intercepting embeddings and **malicious clients**, as detailed in Appendix A.3, though VFL involves other security threats like backdoor and inference attacks that fall outside our scope.
> 4. For scenarios with non-overlapping user sets requiring secure entity alignment, this represents another research topic in VFL [3,5], and decoupling can be integrated with existing privacy-preserving entity resolution techniques since AT operates independently of the entity alignment phase.
>
> **Typo:**
>
> Thank you for catching the typo. We will correct them and clarify DecVFAL-3-3 and DecVFAL-6-2 indicate the settings of M and N.
>
> [1] LIN, Runqi, YU, Chaojian, HAN, Bo and LIU, Tongliang. On the Over-Memorization During Natural, Robust and Catastrophic Overfitting. In: The Twelfth International Conference on Learning Representations. 2024.
>
> [2] WANG, Yisen, et al. Improving adversarial robustness requires revisiting misclassified examples. In: International conference on learning representations. 2020.
>
> [3] LIU, Yang, et al. Vertical federated learning: Concepts, advances, and challenges. IEEE transactions on knowledge and data engineering, 2024, 36.7: 3615-3634.
>
> [4] CROCE, Francesco, et al. RobustBench: a standardized adversarial robustness benchmark. In: Thirty-fifth Conference on Neural Information Processing Systems Datasets and Benchmarks Track (Round 2).
>
> [5] HUANG, Lingxiao, et al. Coresets for Vertical Federated Learning: Regularized Linear Regression and $ K $-Means Clustering. Advances in Neural Information Processing Systems, 2022, 35: 29566-29581.
>
> [6] WEI, Kang, et al. Vertical federated learning: Challenges, methodologies and experiments. arXiv preprint arXiv:2202.04309, 2022.
>
> [7] WANG, Ganyu, et al. A unified solution for privacy and communication efficiency in vertical federated learning. Advances in Neural Information Processing Systems, 2023, 36: 13480-13491.

---

> ### Author Response · Authors · 2025-08-04
> **Additional Results on Deeper Networks**
>
> Dear Reviewer 2ipB,
>
>
> For your concern "W3-Deeper model", We are very excited to share **new experimental results** that directly address the concern about model depth. We have completed experiments on CIFAR-10 using **WideResNet-70-16**, a significantly deeper architecture with 70 layers. DecVFAL achieves **comparable robust accuracy** to PGD while providing **2.46× speedup**. The results demonstrate that DecVFAL maintains its effectiveness on deep networks:
>
> | Method   | Clean Acc | FGSM | PGD  | AutoAttack | Time (h) | Speedup |
> |----------|-----------|------|------|------------|----------------|---------|
> | Clean    | **90.69**     | 17.77| 3.34 | 0.24       | 13.56         | -       |
> | PGD      | 87.52     | 66.34| 68.57| 62.75      | 41.37       | 1.0×    |
> | FreeAT   | 90.30     | 54.11| 56.67| 46.78      | 36.17        | 1.14×   |
> | **DecVFAL** | 87.91 | **66.74** | **68.88** | **63.44** | **16.81** | **2.46×** |
>
>
> We hope our rebuttal can **address your concerns**. If you have any questions, we would be **very happy to discuss them with you in depth**.

---

> ### Author Response · Authors · 2025-08-05
> **Follow-up Discussion - Only 2 days left**
>
> Dear Reviewer 2ipB,
>
> Thank you for your detailed review and the specific concerns you raised. We have provided extensive responses addressing novelty, baselines comparison, deeper network evaluation, and communication analysis. We've also included additional experimental results on WideResNet-70-16 as requested.
>
> **With two days left in the discussion phase**, we'd welcome any further thoughts on our responses. This would allow us to address your concerns more thoroughly.
>
>
> Thank you for your careful evaluation.
>
> Best regards,
>
> The authors of Paper 10820

---

> > ### Comment · Reviewer_2ipB · 2025-08-06
> >
> > After considering the rebuttal, I appreciate the extra experiments and clarifications, but the core concerns on marginal novelty, incomplete and partly unfair empirical comparison (I appreciate the response to W5, but I would ask authors to provide experiments with consistant settings across datasets to make it easier to compare, further as in Table 1 of main paper, the computation advantage moving from MNIST to CIFAR-10 reduces from 10x to 3x against PGD requiring extension to more datasets), and limited evidence of benefit at realistic scale/heterogeneity remain. Therefore, I would like to maintain my score.

---

> ### Author Response · Authors · 2025-08-07
> **Response to "Marginal Novelty" Concern - Technical Challenges**
>
> Dear Reviewer 2ipB,
>
> Thank you for your thoughtful engagement with our work. We respectfully address your concern about "marginal novelty" by clarifying our significant technical and theoretical contributions that **extend well beyond incremental improvements**.
>
> ## Fundamentally Different Application of Decoupling
>
> While previous decoupling works focused on parameter updates during standard training, **DecVFAL is the first approach to apply decoupling to adversarial sample generation**—the most computationally intensive component of AT. This represents a fundamental innovation rather than a marginal adaptation.
>
> Our key insight is that adversarial sample generation offers unique advantages for decoupling that are unavailable in standard training. Since model parameters remain frozen during adversarial sample generation, our approach enables **true parallel processing rather than the approximate parallelism of forward propagations**  in previous work. This fundamental difference allows us to achieve substantially greater efficiency gains while maintaining robustness guarantees.
>
> To demonstrate the standalone value of our approach beyond VFL, we evaluated DecVFAL in a centralized (non-VFL) setting with a single client, simulating local AT on CIFAR-10. **Even when applied to standard PGD-based AT, our method achieves significant improvements**:
>
> | Method | Time (h) | PGD-10  | AA  | Speedup |
> |--------|----------|------------------|--------------|---------|
> | PGD-10 | 6.29 | **50.54** | 45.73 | 1.00× |
> | DecVFAL-6-2 | 2.16 | 50.31 | **45.77** | 2.91× |
>
> Moreover, VFL's natural vertical partitioning perfectly aligns with our layer-wise decoupling mechanism, enabling even greater efficiency gains in distributed settings, as confirmed by our experimental results in the paper.
>
> ## Unique Technical Challenges
>
> Applying decoupling to adversarial sample generation in VFL introduces unprecedented challenges not addressed in prior work:
>
> **1. Preserving Complex Optimization Structure**: Unlike standard training, AT involves intricate minimax optimization that alternates between adversarial sample generation and model parameter updates. Our framework carefully maintains this delicate balance while introducing parallelization. Thsi is a non-trivial work that required novel algorithmic design.
>
> **2. Managing Multi-source Approximation Errors**: The distributed nature of VFL compounds approximation challenges significantly. The VFL framework inherently requires communication-efficient techniques, necessitating the introduction of compression and gradient approximation methods that generate **multiple sources of approximation errors**. These accumulated errors pose a substantial risk of degrading the robustness of AT. Our framework innovatively balances gradient quality and computational efficiency through carefully designed approximation strategies that preserve model robustness while achieving substantial speedups.

---

> > ### Author Response · Authors · 2025-08-07
> > **Response to "Marginal Novelty" Concern - Theoretical Challenges**
> >
> > ## Theoretical Challenges
> >
> > **The Core Theoretical Challenge.** Analyzing VFAL convergence presents an unprecedented theoretical challenge due to the **simultaneous interaction of three distinct approximation error sources** within a complex distributed min-max optimization framework. Unlike standard AT or FL that deal with single error sources, VFAL requires handling delayed gradients from our decoupling mechanism, compressed gradients from VFL communication constraints, and estimated gradients from zeroth-order optimization—all while maintaining the **intricate inner-outer loop structure** of AT **across distributed participants**. Each error propagates through different pathways and interacts in non-trivial ways within the distributed min-max optimization structure.
> >
> > **Theoretical Innovation.** We developed a **unified error decomposition framework** that rigorously characterizes how these three error sources interact within the VFAL system. We leverage optimal control theory to analyze the complex interplay between adversarial sample generation (inner loop) and parameter updates (outer loop). Through this formulation, we established **Lemma 1**, transforming temporal delays in our decoupling mechanism into spatially bounded approximation errors. This foundation allowed us to prove that the three error sources can be **independently bounded and linearly combined** without exponential error amplification.
> >
> > **Achieving Optimal Convergence with Error Characterization.** Our **Theorem 1** establishes that DecVFAL maintains the **$O(1/√K)$ convergence rate** identical to standard VFL, demonstrating that our substantial computational acceleration comes at zero theoretical cost. More importantly, we provide **precise characterization of each error component**, with the total approximation error decomposed as $I₁ + I₂ + Ep + Ec + Ez$, where each term has computable bounds. Our analysis reveals the **bias term $O(N \mathcal{M}_K/M)$** that directly captures the trade-off between modularity and performance, while the **$π(M,N)$ function** provides explicit guidance for optimal parameter selection. This theoretical framework not only proves correctness but delivers **actionable design principles** for practitioners.
> >
> > We believe these contributions constitute significant novelty and technical advancement that go beyond merely applying known techniques to a new setting. Our approach fundamentally changes how AT is conducted in distributed settings, enabling practical deployment of robust VFL systems.
> >
> > Respectfully,
> >
> > The authors of Paper 10820

---

> > > ### Author Response · Authors · 2025-08-07
> > > **Response to "Incomplete and partly unfair empirical comparison"**
> > >
> > > Thank you for this important feedback. We would like to clarify our experimental design and provide additional evidence.
> > >
> > > ## Addressing the Experimental Setting Concerns
> > >
> > > We understand the reviewer's concern about consistency across datasets. However, we **respectfully clarify** that our experimental configurations follow well-established practices in AT literature, as evidenced in benchmark frameworks like PGD, YOPO, TRADES, and RobustBench.
> > >
> > > The different iteration settings between MNIST (40 iterations) and CIFAR (10 iterations) reflect fundamental dataset characteristics rather than arbitrary choices:
> > >
> > > - **MNIST**: The simpler feature space requires more iterations to generate sufficiently challenging adversarial examples
> > > - **CIFAR**: The higher-dimensional, more complex feature space enables effective perturbations with fewer iterations
> > >
> > > Crucially, **all baseline methods use identical dataset-specific configurations** within each experimental setting, ensuring fair comparison with baselines.
> > >
> > > ## Additional Experiments with Consistent Settings
> > >
> > > To directly address your concern, we have conducted supplementary experiments on **MNIST using 10 iterations (consistent with CIFAR settings)**. The following table presents accuracy achieved at fixed time intervals, demonstrating DecVFAL-6-2's efficiency advantages:
> > >
> > > | Algorithm (time/epoch) | 100s | 200s | 500s | 800s | 1000s | 1500s | 2000s | 2500s | 3000s |
> > > |:----------------------:|:----:|:----:|:----:|:----:|:-----:|:-----:|:-----:|:-----:|:-----:|
> > > | PGD-10 (31.62s)        | 85.76% (3) | 90.00% (6) | 93.49% (15) | 94.28% (27) | 94.97% (32) | 95.52% (47) | 95.90% (63) | 96.10% (79) | 95.90% (95) |
> > > | FreeAT-8 (26.05s)      | 87.47% (4) | 91.95% (8) | 94.69% (19) | 95.35% (31) | 95.64% (38) | 96.04% (58) | 96.07% (77) | 96.07% (96) | 96.07% (100) |
> > > | FreeLB-10 (28.94s)     | 80.07% (3) | 84.48% (7) | 87.68% (17) | 85.20% (29) | 81.82% (35) | 77.57% (52) | 75.39% (69) | 75.39% (86) | 75.39% (100) |
> > > | YOPO-5-3 (19.91s)      | 89.82% (5) | 93.92% (10) | 95.40% (25) | 95.61% (40) | 95.43% (50) | 95.68% (75) | 95.69% (100) | - | - |
> > > | **DecVFAL-6-2 (8.76s)** | **90.82% (11)** | **94.27% (23)** | **96.01% (57)** | **96.34% (91)** | **96.34% (100)** | - | - | - | - |
> > >
> > > *Numbers in parentheses indicate completed epochs at each time point.*
> > >
> > > The results demonstrate the significant advantages of DecVFAL-6-2. DecVFAL-6-2 achieves 96% accuracy within 500 seconds, while competing methods require 2-6× longer training time for comparable performance. With 8.76 seconds per epoch, DecVFAL-6-2 completes 100 epochs in 876 seconds, representing a 2.3-3.6× speedup over alternatives while achieving superior final accuracy. At every measured time point, DecVFAL-6-2 outperforms all baselines, confirming robust performance across the entire training trajectory.
> > >
> > > These additional experiments with consistent iteration settings across datasets validate our original findings and demonstrate that DecVFAL-6-2's advantages stem from algorithmic improvements rather than experimental configuration differences. The consistent performance gains across different settings confirm the method's effectiveness and practical value.
> > >
> > >
> > > Respectfully,
> > >
> > > The authors of Paper 10820

---

> > > > ### Comment · Reviewer_2ipB · 2025-08-07
> > > >
> > > > Thank you for providing additional results and clarifying my doubts. I am happy to increase my score.

---

> ### Author Response · Authors · 2025-08-07
> **Response to Reviewer 2ipB's Comment**
>
> Dear Reviewer 2ipB,
>
> Thank you for your positive response and for increasing your score. Your constructive feedback has been invaluable in strengthening our work, and we appreciate your thorough engagement throughout this review process.
>
> Should you have any further questions or concerns, we would be more than happy to address them. We wish you continued success in your research endeavors.
>
> Best regards,
>
> The authors of Paper 10820

---

### Official Review · Reviewer_yT3W · 2025-07-02

**Clarity:** 3
**Significance:** 3
**Originality:** 3
**Rating:** 4
**Confidence:** 3

**Summary:**

This paper introduces DecVFAL, a novel framework that addresses the computational efficiency challenges in VFAL through an innovative dual-level decoupling mechanism. Specifically, the framework first employs "lazy sequential backpropagation" to decouple the client bottom module from the rest of the network, reducing communication frequency between participants; second, it implements "decoupled parallel backpropagation" to enable asynchronous computation across modules, eliminating idle waiting time. The authors provide the first convergence analysis for VFAL, proving that DecVFAL achieves an O(1/√K) convergence rate matching standard VFL. Experimental results demonstrate that DecVFAL achieves 3-10x training acceleration compared to existing methods while maintaining model robustness.

**Questions:**

See above.

**Ethical Concerns:**

["NO or VERY MINOR ethics concerns only"]

**Limitations:**

See above.

**Quality:**

3

**Strengths And Weaknesses:**

## Strengths

1. The paper addresses a critical computational bottleneck in VFAL, which has significant implications for practical deployment. The proposed dual-level decoupling mechanism represents a novel approach that cleverly resolves the inherent sequential dependencies in VFAL.

2. The paper provides the first convergence analysis for VFAL, considering the interaction of multi-source approximation gradients, decoupling mechanisms, and VFL architecture, which is a significant contribution to the theoretical foundation.

3. The work includes extensive evaluations across multiple datasets, different model architectures, and various attack types, demonstrating the effectiveness and generalizability of the method. The detailed analysis of key parameters (M, N), number of modules, and split positions provides valuable insights into the method's design principles.

## Limitations and Questions

1. While the paper mentions differential privacy guarantees, there is limited analysis of whether the decoupling mechanism (particularly delayed gradients) introduces new privacy vulnerabilities. Could malicious participants extract additional information from these delayed gradients?

2. The proposed decoupling strategies are primarily based on feed-forward network architectures. How would these strategies adapt to models with complex structures (e.g., Transformers or networks with dense skip connections), particularly when these structures are distributed across different participants?

3.  While DecVFAL reduces the number of communication rounds, each round may transmit more information. How does this trade-off affect overall performance in bandwidth-constrained environments?

4. The experiments indicate that M and N parameter choices significantly impact performance. Is it possible to develop adaptive strategies that dynamically adjust these parameters based on task characteristics, communication costs, and computational resources?

---

> ### Author Rebuttal · Authors · 2025-07-31
>
> We sincerely thank you for your insightful comments and constructive criticisms. Your feedback has been invaluable in improving the quality and clarity of our manuscript. Below, we address the weaknesses and respond to your questions.
>
> **W1-Privacy Analysis:**
>
> 1. **Delayed gradients do not introduce new privacy vulnerabilities.**
> Delayed gradients transmit same content as immediate gradients differing only in transmission timing. Malicious participants cannot extract additional private information from delayed gradients beyond what they could already obtain from immediate gradients.
>
> 2. DecVFAL operates within the VFL-CZOFO framework [1], which provides differential privacy through zeroth-order optimization. The delayed gradients undergo **the same privacy-preserving transformations as standard gradients**, maintaining equivalent protection levels.
>
> 3. While AT increases communication rounds and **may require a larger privacy budget** compared to standard VFL-CZOFO, we will provide detailed privacy analysis in the revised version to quantify this trade-off between robustness and privacy.
>
> **W2-Model with Complex Architectures:**
>
> Decoupling easily adapts to complex architectures by treating functional blocks as module boundaries [2-6].
> For complex structures like Transformers and residual networks, we can treat entire functional blocks (including their attention or residual connections) as **single dynamical units**. This preserves the sequential flow while enabling parallel processing. Our ResNet-18 experiments illustrate this concept: residual blocks naturally work as complete dynamical units, maintaining both the residual connections and the dynamical systems framework (footnote 4, page 7).
>
>
> **W3-Transmitted Content:**
>
> DecVFAL and standard VFAL transmit **the same content** (i.e. forward propogation for embeddings and backward propagation for  gradients).DecVFAL achieves efficiency by reusing gradients to reduce the total number of communication rounds.
>
> **W4-The selection of M and N:**
> 1. The optimal M and N values depend on many factors: dataset size and features, model structure, and training settings like learning rate and adversarial step size. Because of these complex interactions, developing an adaptive strategy is difficult.
> 3. **Practical Selection Strategy.** Empirically, M and N can be selected by referencing the choice of r in PGD-r, where **M×N should be slightly larger than r**, and maximizing M within the communication budget typically achieves excellent performance (only M relates to server-client communication).
>
>
>
> [1] WANG, Ganyu, et al. A unified solution for privacy and communication efficiency in vertical federated learning. Advances in Neural Information Processing Systems, 2023, 36: 13480-13491.
>
> [2] CHEN, Ricky TQ, et al. Neural ordinary differential equations. Advances in neural information processing systems, 2018, 31.
>
> [3] LI, Qianxiao, et al. Maximum principle based algorithms for deep learning. Journal of Machine Learning Research, 2018, 18.165: 1-29.
>
> [4] LI, Qianxiao; HAO, Shuji. An optimal control approach to deep learning and applications to discrete-weight neural networks. In: International Conference on Machine Learning. PMLR, 2018. p. 2985-2994.
>
> [5] HAN, Jiequn, et al. A mean-field optimal control formulation of deep learning. Research in the Mathematical Sciences, 2019, 6.1: 1-41.
>
> [6] WEINAN, Ee. A proposal on machine learning via dynamical systems. Communications in Mathematics and Statistics, 2017, 5.1: 1-11.

---

> ### Author Response · Authors · 2025-08-05
> **Follow-up Discussion - Only 2 days left**
>
> Dear Reviewer yT3W,
>
> We appreciate your insightful review and the important questions you raised about privacy analysis, complex architectures, and parameter selection. Our rebuttal provides detailed responses to each of your concerns.
>
> We would welcome any additional feedback or clarifications you might need, as this would help us ensure a comprehensive discussion of our work.  Any additional feedback in the remaining two days would be much appreciated.
>
> Thank you for your thoughtful evaluation.
>
> Best regards,
>
> The authors of Paper 10820

---

### Official Review · Reviewer_ikWZ · 2025-07-06

**Clarity:** 3
**Significance:** 3
**Originality:** 3
**Rating:** 4
**Confidence:** 3

**Summary:**

DecVFAL accelerates adversarial training in VFL via dual-level decoupling (lazy sequential + parallel backpropagation), achieving 3–10× speedup with rigorous convergence guarantees and competitive robustness.

**Questions:**

1. While experiments on public datasets (MNIST/CIFAR) demonstrate efficiency, how does DecVFAL perform on industry VFL benchmarks (e.g., financial fraud detection or healthcare datasets with heterogeneous feature distributions)?

2. Corollary 1 identifies a bias term \( \mathcal{O}(N\mathcal{M}_K/M) \) that degrades performance with increasing modules  \mathcal{M}_K . What practical strategies can mitigate this?

3. DecVFAL assumes synchronous client-server updates (Algorithm 1). How would asynchronous settings (common in cross-device VFL [36]) affect convergence?

**Ethical Concerns:**

["NO or VERY MINOR ethics concerns only"]

**Limitations:**

See questions

**Quality:**

3

**Strengths And Weaknesses:**

Strengths:​​

​​1. Rigorous Technical Foundation​​: Proposes a mathematically grounded dual-decoupling mechanism (lazy sequential + parallel backpropagation) with convergence guarantees (Theorem 1, Corollary 1).
2. ​​Comprehensive Evaluation​​: Validated across 4 datasets (MNIST, CIFAR-10/100, Tiny-ImageNet), 8 baselines (PGD, FreeAT, YOPO), and 10 attack methods (FGSM, PGD, CW, etc.), demonstrating ​​3–10× speedup​​ without compromising robustness (Tables 1–3).
3. ​​Theoretical-Experimental Alignment​​: Convergence analysis explicitly accounts for VFL-specific challenges (e.g., gradient delays, compression errors) and aligns with empirical results.

Weaknesses:

​​1. Limited Real-World Data​​: Experiments use partitioned public datasets (Sec. 6.1). Testing on industry VFL benchmarks (e.g., finance/healthcare) would strengthen practicality.
2. Bias Term Trade-Off: Convergence rate includes \( \mathcal{O}(N\mathcal{M}_K/M) \) (Corollary 1), implying performance degradation with many modules. Not empirically explored beyond Table 5.
3. Decoupling Inspiration: Builds on prior work (e.g., Synthetic Gradients [27], ADMM [57]), though VFAL-specific extensions are non-trivial.

---

> ### Author Rebuttal · Authors · 2025-07-31
>
> We sincerely appreciate your insightful comments and constructive feedback, which have been invaluable in improving the quality and clarity of our manuscript. Below, we provide detailed responses to the identified weaknesses and questions.
>
> **W1&Q1-Real-World Dataset:**
>
> Thank you for the recognition of this limitation. To address your concern, we added **2 real-world datasets: COVID-19 Image Data Collection [1] and Credit Card Fraud Detection [5]** to further validate our work. **VFL Setup:** we set 2 clients which each uses ResNet-18 for COVID-19 and MLP for Credit, one server employs 1-layer classifier. DecVFAL achieves **3.72× speedup** and **2.35× speedup** respectively compared to standard PGD, while maintaining superior robustness.
>
>
> Table 1: Results of Robust Training on **COVID-19 dataset [1]**
>
> | Methods  | clean  | FGSM  | PGD   | AA    | Time (min) &darr; | Speedup (vs PGD) |
> |---------|-------|-------|-------|-------|------------|------------------|
> | None   | **98.19** | 0.36  | 0.00  | 0.00  | -       | -            |
> | PGD     |  69.20|  67.02|  **67.39**|  **66.30** | 29.64    | 1.00×            |
> | FreeAT  | 77.53 | 49.27 | 50.54 | 40.76  |    11.65   | 2.54×            |
> | YOPO  |  77.90 | 66.30 | 66.49 | 58.51  |   8.73    |3.39×            |
> | DecVFAL |  88.22 |  **67.39**|  **62.86**|  44.75|  7.97     | **3.72×**            |
>
> Table 2: Results of Robust Training on **Credit dataset [5]**
>
> | Methods  | clean | FGSM  | PGD   | AA    | Time (min) &darr;| Speedup (vs PGD) |
> |---------|-------|-------|-------|-------|------------|------------------|
> | None   | **92.45** | 0.00  | 2.86  | 3.38  | 5.19       | -            |
> | PGD     | 88.28 | 40.63 | 42.19 | **49.48** | 36.02      | 1.00×            |
> | FreeAT  | 92.44 | 14.58 | 40.36 | 5.72  | 27.16      | 1.33×            |
> | YOPO | 90.10 | 54.68 | 32.29 | 22.39 | 17.98 | 2.00× |
> | DecVFAL | 89.58 | **68.22** | **47.13** | 47.39 | **15.32**      | **2.35×**            |
>
> **W2&Q2-Impact of the number of modules:**
>
> 1. The trade-off between module number and accuracy is inherent to module decoupling and must be carefully balanced. Our **theoretical analysis (Remark 1)** and **empirical results (Table 5)** consistently demonstrate that increasing module numbers $\mathcal{M}_K$ degrades performance. This outcome **is consistent with prior decoupled training work** [2-4].
>
> 2. The practical strategies to mitigate this include (1) against excessive module partitioning, (2) align the execution time of all partitioned modules to maximize parallelization benefits.
>
>
> **W3-Novelty:**
>
>
> Thank you for recognizing the non-trivial contributions of our framework. We want to emphasize the unique challenges our work addresses.
>
> 1. **Technical Innovation:** While we build upon fundamental techniques, applying decoupling to communication-efficient VFL presents distinct challenges. Our approach uniquely combines decoupling with AT and VFL-specific methods like Zero-Order Optimization and compression [8]. The key innovation lies in **decoupling adversarial sample generation with frozen parameters** - a departure from prior work [2-4] that only decouples parameter updates. This enables true concurrent forward/backward propagation and addresses the computational bottleneck in adversarial sample generation, achieving substantial 3-10× acceleration for VFAL.
>
> 2. **Theoretical Contribution:** The simultaneous presence of **multiple gradient approximations** creates analytical complexity not addressed in existing literature. Our framework must account for three interacting approximation sources: delayed gradients (from decoupling), compressed gradients (from embedding compression), and estimated gradients (from ZOO). We provide the first convergence analysis that captures how these approximation errors propagate and interact within VFAL systems, requiring careful treatment of their cumulative effects.
>
>
> **Q3-Asynchronous Settings Impact:**
>
> We think that full asynchronous VFL updating presents significant convergence challenges. AT operates through **a minimax optimization** that alternates between adversarial sample generation and model parameter updates. This requires **freezing model parameters during adversarial sample generation**, followed by synchronized parameter updates using the generated samples. Asynchronous settings would **disrupt this minimax execution order**, potentially leading to scenarios where model parameters are updated before optimal adversarial samples are identified.
>
>
>
> [1] MAGUOLO, Gianluca; NANNI, Loris. A critic evaluation of methods for COVID-19 automatic detection from X-ray images. Information fusion, 2021, 76: 1-7.
>
> [2] JADERBERG, Max, et al. Decoupled neural interfaces using synthetic gradients. In: International conference on machine learning. PMLR, 2017. p. 1627-1635.
>
> [3] HUO, Zhouyuan, et al. Decoupled parallel backpropagation with convergence guarantee. In: International Conference on Machine Learning. PMLR, 2018. p. 2098-2106.
>
> [4] HUO, Zhouyuan; GU, Bin; HUANG, Heng. Training neural networks using features replay. Advances in Neural Information Processing Systems, 2018, 31.
>
> [5] CARCILLO, Fabrizio, et al. Combining unsupervised and supervised learning in credit card fraud detection. Information sciences, 2021, 557: 317-331.
>
>
> [6] CROCE, Francesco, et al. RobustBench: a standardized adversarial robustness benchmark. In: Thirty-fifth Conference on Neural Information Processing Systems Datasets and Benchmarks Track (Round 2).
>
> [7] LIU, Yang, et al. Vertical federated learning: Concepts, advances, and challenges. IEEE transactions on knowledge and data engineering, 2024, 36.7: 3615-3634.
>
> [8] WANG, Ganyu, et al. A unified solution for privacy and communication efficiency in vertical federated learning. Advances in Neural Information Processing Systems, 2023, 36: 13480-13491.

---

> ### Author Response · Authors · 2025-08-05
> **Follow-up Discussion - Only 2 days left**
>
> Dear Reviewer ikWZ,
>
> Thank you for your thorough review and constructive feedback on our paper. We have addressed your concerns regarding real-world datasets, bias term trade-offs, and asynchronous settings in our detailed rebuttal.
>
> With the discussion phase ending in two days, we'd greatly appreciate any follow-up comments you might have.
>
> Thank you for your time and valuable insights.
>
> Best regards,
>
> The authors of Paper 10820

---

### Author Response · Authors · 2025-08-09
**Final Acknowledgment to Area Chairs and Reviewers**

Dear ACs and Reviewers,

As the rebuttal period is concluding for Paper 10820: "Accelerated Vertical Federated Adversarial Learning through Decoupling Layer-Wise Dependencies," we sincerely thank you for your thorough reviews and constructive feedback. To facilitate your final evaluation, we summarize our rebuttal responses below:

## Key Contributions

Our DecVFAL framework introduces the **first application of decoupling to adversarial sample generation**, fundamentally different from prior work that focused on parameter updates. This innovation enables true parallel processing with frozen parameters, achieving **3-10× speedup while maintaining O(1/√K) convergence rate**. We provide the **first theoretical analysis for VFAL**, rigorously characterizing the interaction of three approximation sources (delayed, compressed, estimated gradients) within distributed adversarial training.

## Response to Main Reviewer Concerns

- **Novelty (Reviewer 2ipB)**: We clarified that decoupling adversarial sample generation presents fundamentally different challenges compared to parameter update decoupling. Additional centralized experiments demonstrated 2.91× speedup even without VFL, proving standalone algorithmic value.

- **Real-world Evaluation (Reviewers ikWZ, yT3W)**: We added COVID-19 medical imaging (3.72× speedup) and credit card fraud detection (2.35× speedup) datasets, validating performance in industry-relevant VFL scenarios beyond public benchmarks.

- **Architecture Scalability (Reviewer 2ipB)**: Extended validation to WideResNet-70-16 with 70 layers, achieving 2.46× speedup. Additional analysis of server model complexity (1/4/16-layer configurations) demonstrated architectural flexibility.

- **Client Scale (Reviewers 2ipB, Ecn8)**: Expanded experiments to 14 clients (4.35× speedup) and 28 clients (4.92× speedup) on MNIST, maintaining consistent robustness across larger federated scenarios.

- **Experimental Rigor (Reviewer 2ipB)**: Provided consistent iteration settings across datasets, integrated recent baselines (DOM, MART) with competitive results, and quantified communication costs.

- **Theoretical Depth (Reviewers ikWZ, yT3W)**: Characterized module number trade-offs with practical mitigation strategies, confirmed equivalent privacy preservation to standard VFL, and provided explicit parameter selection guidance.

DecVFAL addresses the critical computational bottleneck in VFAL by dramatically reducing adversarial training time. Our expanded experimental validation across real-world datasets, deep architectures, and large-scale scenarios demonstrates broad practical applicability. The rigorous theoretical framework provides both correctness guarantees and actionable design principles for users.

We deeply appreciate the constructive scholarly discussion, which has been invaluable for improving our work.


Best regards,

The Authors of Paper 10820

---

### Note · Authors · 2025-08-15

Dear ACs and Reviewers,

We sincerely thank the ACs and reviewers for their thorough feedback and coordination throughout the rebuttal and discussion process. To facilitate the final evaluation, we summarize below:

**Contributions Clarified:**

1. **Technical Innovation:** We introduce the **first application of decoupling to adversarial sample generation of AT**, enabling true parallel propagations with frozen parameters. Even in centralized settings, our method achieves significant speedup. VFL's vertical partitioning naturally aligns with our layer-wise decoupling for greater distributed efficiency in practice.
2. **Theoretical Challenge:** Analyzing VFAL convergence presents unique challenges due to **multiple approximation error interactions** within **distributed min-max optimization** while maintaining AT's intricate inner-outer loop structure across participants. We provide convergence rate matching standard VFLs with **precise error characterization**, delivering actionable design principles.
3. **Comprehensive Evaluation:** Validation across 6 datasets (4 benchmark + 2 realistic), multiple baselines and attack methods. **Extensive ablations** on module numbers, M and N parameters, split positions, and client scaling fully demonstrate significant speedup without compromising robustness.

**Rebuttal Summary:**

- **Reviewers ikWZ and yT3W** provided **positive scores** and raised few concerns that we believe our detailed rebuttals, including additional experiments, clearly addressed, though they did not provide follow-up responses.
- **Reviewer 2ipB** engaged in multiple rounds of productive discussion and explicitly stated: "Thank you for providing additional results and clarifying my doubts. I am happy to **increase my score**."
- **Reviewer Ecn8** provided **positive score** and raised few concerns and confirmed that "concerns have been addressed."
- While reviewers confirmed their concerns were addressed, we'd like to clarify 2 minor points: (1) heterogeneity represents a separate VFL research challenge beyond AT scope; (2) in VFL's **feature-partitioned** framework, excessively large client numbers are unrealistic (our experiments spanning 7~28 clients already demonstrate comprehensive scalability).

In summary, DecVFAL addresses the critical computational bottleneck in AT in distributed settings, providing both VFAL theoretical analysis frameworks and significant acceleration solutions in practice.

Best regards,

The Authors of Paper 10820

---

### Decision · Program_Chairs · 2025-09-17

**Decision:**

Accept (poster)

**Comment:**

This paper proposes a novel method which accelerates the adversarial training process in Vertical Federated Learning (VFL) by introducing a decoupling strategy with lazy sequential back-propagation, achieving significant speedup in performance. It also provides rigorous convergence guarantees and competitive robustness.  One weakness is the marginal performance gain for bigger datasets, as pointed by reviewer 2ipB.

During the rebuttal process, the authors addressed most of the concerns from the reviewers, and no additional concerns is raised. After the AC-reviewer discussion, the reviewers reached a consensus that the paper can be accepted.

Considering the consensus of reviewers and the nontrivial technical contribution of this work targeting a timely and important challenge,  efficiency bottleneck of VFL, I recommend the acceptance of this paper.